# Assessment of the peak tsunami amplitude associated with a great earthquake occurring along the southernmost Ryukyu subduction zone in the region of Taiwan

Yu-Sheng Sun[1], Po-Fei Chen[1], Chien-Chih Chen[1,2], Ya-Ting Lee[1,2], Kuo-Fong Ma[1,2] and Tso-Ren Wu[2,3]

[1]Department of Earth Sciences, National Central University, Taoyuan City 32001, Taiwan, R.O.C.
[2]Earthquake-Disaster & Risk Evaluation and Management Center, National Central University, Taoyuan City 32001, Taiwan, R.O.C.
[3]Graduate Institute of Hydrological and Oceanic Sciences, National Central University, Taoyuan City 32001, Taiwan, R.O.C.

*Correspondence to*: Yu-Sheng Sun (shengfantasy@gmail.com)

**Abstract.** The southernmost portion of the Ryukyu Trench near the island of Taiwan potentially generates tsunamigenic earthquakes with magnitudes from 7.5 to 8.7 through shallow rupture. The fault model for this potential region dips 10° northward with a rupture length of 120 km and a width of 70 km. An earthquake magnitude of Mw 8.15 is estimated by the fault geometry with an average slip of 8.25 m as a constraint on the earthquake scenario. Heterogeneous slip distributions over the rupture surface are generated by a stochastic slip model, which represents that the slip spectrum decays according to $k^{-2}$ in the wavenumber domain. These synthetic slip distributions are consistent with the above mentioned identical seismic conditions. The results from tsunami simulations illustrate that the propagation of tsunami waves and the peak wave heights largely vary in response to the slip distribution. Changes in the wave phase are possible as the waves propagate, even under the same seismic conditions. The tsunami energy path not only follows the bathymetry but also depends on the slip distribution. The probabilistic distributions of the peak tsunami amplitude calculated by 100 different slip patterns from 30 recording stations reveal that the uncertainty decreases with increasing distance from the tsunami source. The highest wave amplitude for 30 recording points is 7.32 m at Hualien for 100 different slips. Compared with the stochastic slip distributions, the uniform slip distribution will be extremely underestimated, especially in the near field. In general, the uniform slip assumption represents only the average phenomenon and will consequently ignore the possibility of tsunami waves. These results indicate that considering the effects of heterogeneous slip distributions is necessary for assessing tsunami hazards to provide additional information about tsunami uncertainties and facilitate a more comprehensive estimation.

## 1 Introduction

Almost all destructive tsunamis are generated by shallow earthquakes that occur within subduction zones. Numerous destructive tsunami events, including the Mw 9.1 Sumatra earthquake in 2004 (Lay et al., 2005), the Mw 8.8 Chile earthquake in 2010 (Lay et al., 2010; Fritz et al., 2011) and the Mw 9.0 Tohoku earthquake in 2011 (Goda et al., 2015; Goda and Song, 2016), all of which occurred in subduction zones, have occurred recently. The island of Taiwan, which is located at the

convergent boundary between the Philippine Sea Plate and the Eurasian Plate, is constantly under the possible threat of a tsunami. The convergence rate in this area is approximately 80-85 mm/yr (Seno et al., 1993; Yu et al., 1997; Sella et al., 2002; Hsu et al., 2009; Hsu et al., 2012). Thus, earthquakes occur frequently in and around Taiwan. The shallow earthquakes that occur in the Manila Trench to the south and the Ryukyu Trench to the northeast are particularly tsunamigenic, and earthquakes

occur more actively in the southernmost Ryukyu Trench than in the northern Manila Trench (Wu et al., 2013). The most well-known historic tsunami events that have occurred in northeastern Taiwan are the 1867 Keelung earthquake (Mw 7.0) (Tsai, 1985; Ma and Lee, 1997; Cheng et al., 2016; Yu et al., 2016) and the 1771 Yaeyama (Japan) earthquake ($M_w$~8) (Nakamura, 2009a). Accordingly, these historic recordings demonstrate that Taiwan is under a potential tsunami threat. Furthermore, the 2011 Tohoku earthquake induced a powerful tsunami that destroyed coastal areas and caused nuclear accidents (Mimura et al.,

2011). As there are four nuclear power plants along the coast of Taiwan, it is necessary to carefully estimate the tsunami hazard in addition the hazards of compound disasters.

Probabilistic tsunami hazard analysis (PTHA) is a modification of probabilistic seismic hazard analysis (PSHA) (Cornell, 1968; SSHAC, 1997), and it is intended to forecast the probability of tsunami hazards for a given region as comprehensively as

possible. The recurrence rates of earthquakes have typically been estimated using the Gutenberg–Richter relationship (Gutenberg and Richter, 1944) for a defined source region in consideration of tsunamis triggered by earthquakes. The assessment of the wave height is one of the primary differences between PTHA and PSHA. PSHA assesses the ground motion based on empirical attenuation relationships (Wang et al., 2016), while PTHA assesses tsunami wave heights using empirical approaches or tsunami simulations (Geist, 2002; Geist and Parsons, 2006; Geist and Parsons, 2009). Geist and Parsons (2006)

mentioned that the tsunami wave height follows a definable frequency-size distribution over a sufficiently long period of time within a given coastal region (Soloviev, 1969; Houston et al., 1977; Horikawa and Shuto, 1983; Burroughs and Tebbens, 2005). This method is of great use in establishing the tsunami probability for a region if there is an extensive catalog of observed tsunami wave heights. However, given the wide distribution of global tsunamigenic earthquakes within seafloor regions throughout subduction zones, the tsunami records obtained from coastal gauges or/and ocean buoys are too sparse to

comprehensively assess the associated hazards, and the recording time since their deployment is too short to enable a study of the recurrence intervals of tsunamis/earthquakes. Consequently, because the existing tsunami catalogue is limited, simulations represent an effective approach. Conventional tsunami simulation adopts a simple source approximation and applies elastic dislocation theory to calculate the deformation of the seafloor surface assuming a uniform slip over the entire fault surface (Okada, 1985; Okal, 1982). However, the complexities of earthquake rupture processes play a substantial role in the generation

of tsunamis. Conventional approaches are therefore unable to capture various features of short-wavelength tsunamis in the near field (Geist, 2002; Geist and Parsons, 2009). The results of previous studies that simulated tsunamis originating from historical earthquakes around Taiwan (Ma and Lee, 1997; Wu et al., 2008) using uniform slip models agreed only with long-wavelength observations. For the purposes of hazard mitigation, it is critical to predict the amplitudes of tsunamis along various coastlines for a given earthquake as accurately as possible. To make such predictions, the effects of the rupture complexity

must be taken into consideration. Recent developments in PTHA have included the adoption of stochastic slip distributions of earthquakes to determine the overall probability of particular tsunami heights (Geist and Parsons, 2006, 2009). The adoption of stochastic slip distributions is able to quantify the variations in reasonable evaluations of the probabilities of specified tsunami heights at individual locations resulting from a specific fault.

In this study, we assess the heights of tsunamis along the coastline of Taiwan generated by the potential tsunamigenic zone at the southernmost end of the Ryukyu subduction zone. This potential zone is located close to Taiwan, and at least ten earthquakes ($M_w$>7) have occurred over the past 100 years (Hsu et al., 2012), the largest of which was the Mw 7.7 in 1920 (Theunissen et al., 2010). For this area, the plausible magnitude of greatest earthquake is determined within the range between

10    7.5 and 8.7 ($M_w$) (Hsu et al., 2012). The fault zone is bounded by the Longitudinal Valley Fault to the west and the Gagua Ridge to the east (Hsu et al., 2012). This fault geometry with a defined rupture length and width is employed herein, and an earthquake with a magnitude of 8.15 is used in the tsunami simulation. The stochastic slip model is invoked to describe the uncertainty in the rupture pattern over the fault plane to enable a more realistic assessment of the tsunami probability.

## 2 Earthquake scenario and tsunami simulation

### 2.1 Assessment of Seismic Parameters

The estimated maximum magnitude of a possible earthquake scenario is essential for establishing the fundamental seismic conditions of the tsunami simulation. The scenario of a potential rupture fault extending to a depth of 13 km proposed by Hsu et al. (2012) occurs along the southernmost Ryukyu trench with a rupture length of 120 km, a width of 70 km and a dip of 10°.

20    Kanamori and Anderson (1975) investigated the relation between the rupture area and moment and revealed that most of the average stress drops ($\Delta\sigma$) vary between 10 and 100 bars. The average stress drop for most interplate earthquakes is approximately 30 bars, and thus, we set an average stress drop of 30 bars. The stress drop and seismic moment ($M_0$) relation along a dip slip fault is described as follows (Kanamori and Anderson, 1975):

$$M_0 = \frac{\pi(\lambda+2\mu)}{4(\lambda+\mu)}\Delta\sigma W^2 L \tag{1}$$

25    where $W$ and $L$ are the width and length of the rupture plane, respectively. We can obtain the moment for this scenario under an average stress drop of 30 bars with the assumed rupture geometry. In Eq. (1), $\mu$ denotes the rigidity and $\lambda$ is the Lamè parameter. We assume that the crust is elastic and homogeneous. Hence, $\mu=\lambda=30$ GPa (Fowler, 2004; Piombo et al, 2007). Additionally, the seismic moment can be represented by the rupture area and average slip as follows (Lay and Wallace, 1995):

$$M_0 = \mu A \overline{D} \tag{2}$$

Moreover, the seismic moment is dependent on the rupture area ($A$) and average slip ($\overline{D}$); thus, the average slip can be estimated by Eq. (2), and it is calculated to be 8.25 m. Then, the seismic moment can be transformed into the magnitude $M_{\mathrm{w}}$ by the following (Hanks and Kanamori, 1979):

$$M_{\mathrm{w}} = \left(\frac{\log M_0}{1.5}\right) - 10.73 \tag{3}$$

Therefore, the maximum possible earthquake magnitude is $M_{\mathrm{w}}$ 8.15 ($M_0 = 2.07 \times 10^{28}$ dyne-cm).

## 2.2 Stochastic Slip Model

The rupture process of an earthquake is extremely complex. Seismic inversion results reveal that the slip distribution of a rupture has a heterogeneous spatio-temporal development. Consequently, using a simplified uniform slip distribution to
simulate a tsunami captures only the long-wavelength portion of the tsunami field (Geist and Dmowska, 1999). In addition, the temporal description of the seismic rupture process can be ignored because the propagation velocity of the tsunami wave is substantially slower than the seismic rupture velocity (Dean and Dalrymple, 1991; Ma et al., 1991; Wang and Liu, 2006). Andrews (1980) showed that the static slip distribution is directly related to stress changes and that the spectrum of the slip distribution is proportional to $k^{-2}$ decay in the wavenumber domain:

$$\left|F_{s,t}\left[D_{x,y}\right]\right| \propto k^{-2} \tag{4}$$

where $D_{x,y}$ is the slip distribution over a 2D lattice, $F_{s,t}$ is the 2D Fourier transform, and $k = \sqrt{k_x^2 + k_y^2}$ is the radial wavenumber. The $k^{-2}$ power law indicates that the slip distribution has self-similar characteristics; moreover, this characteristic can also be demonstrated from a fractal perspective (Tsai, 1997). Based on self-similarity, Herrero and Bernard (1994) introduced the $k$-square model, which leads to the ω-square model (Aki, 1967). The slip spectrum follows $k^{-2}$ decay beyond
the corner radial wavenumber ($k_{\mathrm{c}}$), which is proportional to $1/L_{\mathrm{c}}$. The $L_{\mathrm{c}}$ depends on the characteristic rupture dimension (Geist, 2002).

The heterogeneous slip distribution is proportional to $k^{-2}$ and is similar to fractional Brownian motion as a stochastic process (Tsai, 1997). The stochastic slip distribution can be described by multiplication in the Fourier domain:

$$D_{x,y} \propto F_{x,y}^{-1}\left[F_{s,t}\left[X_{x,y}\right] \times k^{-2}\right] \tag{5}$$

where $X_{x,y}$ is a random variable for the spatial distribution that randomizes the phase, and $F_{x,y}^{-1}$ is the inverse 2D Fourier transform. The random distribution of $X$, which is best described by a non-Gaussian distribution, especially by a Lèvy distribution, can be calculated by reversing Eq. (5) (Lavallée and Archuleta, 2003; Lavallée et al., 2006). The Lèvy distribution can be described by four parameters, namely, $\alpha$, $\beta$, $\gamma$ and $\mu_{\mathrm{L}}$, as follows:

$$\varphi(t) = \begin{cases} \exp\left(-\gamma^{\alpha}|t|^{\alpha}\left[1 + i\beta \, \text{sign}(t)\tan\frac{\pi\alpha}{2}(|\gamma t|^{1-\alpha} - 1)\right] + i\mu_L t\right), & \alpha \neq 1 \\ \exp\left(-\gamma|t|\left[1 + i\beta\frac{2}{\pi}\text{sign}(t)(\ln|t| + \ln\gamma)\right] + i\mu_L t\right) & , \; \alpha = 1 \end{cases} \tag{6}$$

The parameter $\alpha$, $0<\alpha\leq2$, affects the falloff rate of the probability density function (PDF) for the tail. The parameter $\beta$, $-1\leq\beta\leq1$, controls the skewness of the PDF, and the parameter $\gamma$, $\gamma>0$, controls the width of the PDF. The parameter $\mu_L$, $-\infty<\mu_L<\infty$, is related to the location of the PDF. The Lèvy distribution is effective at describing the distribution of a random variable, i.e., $X$, from real earthquake events, implying that the slip distribution without self-similarity has a heavy tail behavior (Lavallée et al., 2006). Based on experiments of generating stochastic slip distributions, this heavy tail behavior affects the intensity of an extreme value (Lavallée and Archuleta, 2003).

The stochastic slip distribution is generated by a 2D spatially random distribution by imposing a self-similar characteristic beyond the corner radial wavenumber, which is constrained by the rupture dimension, in the wavenumber domain. In this study, the potential rupture fault is divided into $5\times5$ km$^2$ subfaults. The grid is composed of $24\times14$ meshes along the strike and dip directions, respectively. The produced variable with a spatially random distribution adopts the Lèvy distribution ($\alpha=1.51$, $\beta=0.2$, $\gamma=28.3$, $\mu_L=-0.9$), which is the dip slip result from Lavallée et al. (2006) as shown in Fig. 1a. In Lavallée et al. (2006), the slip distribution of the Northridge earthquake was divided into the dip-slip and strike-slip directions, and they were calculated by an inverse 2D stochastic model to obtain the values of the Lèvy PDF. The values of the Lèvy PDF, which are mentioned above are indicative of the result of dip-slip direction. The Northridge earthquake is a thrust earthquake (Davis, 1994), and thus, it has a faulting mechanism that is approximately similar to our scenario fault model. There are no inverted slip models of past earthquakes in the study area to conduct an analysis of the Levy PDF parameters; therefore, the Lèvy distribution in Lavallée et al. (2006) is adopted in this study. From the perspective of mathematical operations, the slip distribution in Eq. (5) represents a filtered random distribution. However, for consistency with the physical behavior over the rupture surface suggested by the results of the inverse modeling, truncation of the Lèvy distribution must be performed to constrain the extreme slip value. The synthetic slip distribution (Fig. 1b) produced by the spatially random distribution in Fig. 1a is heterogeneous, and its power spectrum obeys a $k$-square model at high wavenumbers (Fig. 1c). The average slip of this synthetic slip distribution is 8.25 m, indicating that the earthquake energy is constant as estimated above, and the maximum slip is 31.02 m. One hundred different slip distributions are produced for the tsunami simulation representing the uncertainty in the results associated with complex rupture processes. In the 100 sets of results, the maximum slip range is between 20.17 and 37.97 m. Smooth processes are not included, nor are additional regional constraints for the slip distribution. There are two reasons for this application. The first is that we do not have information regarding where the plate interface is locked or the locations of asperities often repeat in historical events. The second is that some studies reported that the asperities extend to the boundary of the fault model (Ide et al., 2011; Lay et al., 2011; Shao et al., 2011; Yue and Lay, 2011). According to these reasons, we do not apply any additional constraints for stochastic slip distributions. Similarly, the uniform slip case constitutes a complete uniform slip distribution over the whole fault plane. Figure 1b and 1d demonstrate the stochastic distribution of the

scenario source models causing the maximum and minimum wave heights, respectively, at recording station 26 (Hualien) (Fig. 2). Both patterns affecting the propagation will be discussed in Sect. 3.1.

## 2.3 Numerical Tsunami Simulation

Figure 2 shows the computational domain, recording stations and fault model. The potential rupture fault is divided into 5×5 km$^2$ subfaults, and the stochastic slip distribution model is applied to determine the amount of discrete slip on each subfault. Vertical seafloor displacements caused by slip along the rupture plane are calculated using elastic dislocation theory (Okada, 1985). The Cornell Multi-grid Coupled Tsunami model (COMCOT) is used to perform the tsunami simulations. COMCOT is capable of efficiently studying the entire life-span of a tsunami, including its generation, propagation, runup and inundation

(Wang, 2009), and it has been widely used in studying many historical tsunami events, such as the 1960 Chilean tsunami (Liu et al., 1995), 1992 Flores Islands tsunami (Liu et al., 1995), 2003 Algeria tsunami (Wang and Liu, 2005), 2004 Indian Ocean tsunami (Wang and Liu, 2006, 2007), and 2006 Ping-Tung tsunami, Taiwan (Wu, et al., 2008; Chen, et al., 2008). COMCOT solves linear or nonlinear shallow water equations for spherical or Cartesian coordinates using the finite difference method. With a flexible nested grid system, it can properly guarantee both the efficiency and the accuracy from the near-coastal region

to the far-field region. Two grid layers are used to simulate the propagation of tsunamis. The Manning coefficient is 0.013 in this study to assume a sandy sea bottom (Wu et al., 2008). The bathymetry adopted open data from the National Oceanic and Atmospheric Administration (NOAA) that can be download from https://maps.ngdc.noaa.gov/viewers/wcs-client/ (Amante and Eakins, 2009). The resolution of the outer layer is 4 minutes for the solution of the linear shallow water equation, and the resolution of the inner layer is 1 minute for the solution of the nonlinear form of the shallow water equation. There are 30

recording stations referring to the positions of tidal gauges maintained by the Central Weather Bureau (CWB) along the coastlines of Taiwan and the outlying islands. The CWB website presents the locations of the tide stations (http://e-service.cwb.gov.tw/HistoryDataQuery/index.jsp and http://www.cwb.gov.tw/V7e/climate/marine_stat/tide.htm). These locations are shifted slightly to the grid nodes to accurately record the data. Table 1 presents the locations and water depths of the recording stations in the computational mesh.


## 3 The effect of heterogeneous slip on tsunamis

The stochastic slip model produces different slip distributions with the same fault geometry in addition to a constant average slip and a constant seismic moment. The model is used to describe the heterogeneous slip pattern of an earthquake and to further examine its effect on the tsunamis originating from the southernmost end of the Ryukyu subduction zone adjacent to

Taiwan. According to the previous sections, the maximum possible earthquake magnitude is determined to be $M_w$ 8.15 with

an average slip of 8.25 m. Furthermore, the uniform slip distribution on the rupture plane is also used to simulate tsunami to facilitate a discussion of the difference between the effects of uniform and heterogeneous slip on tsunamis.

### 3.1 Initial water elevation and energy propagation

The static vertical displacement of the ocean floor is modeled using elastic dislocation theory (Okada, 1985) with a static slip distribution. The vertical seafloor displacement is modeled as the initial water level, and the horizontal component of the seabed displacement is not included in the simulation. Figure 3a shows the initial water elevations produced by a uniform slip distribution, and Fig. 3b exhibits the maximum free-surface elevation during the propagation. Figure 3c and 3e demonstrate the initial water elevations produced by the stochastic slip distributions (Fig. 1b and 1d). The initial water elevation with a
uniform slip distribution is simple and smooth, but those with stochastic slip models are more complex and relatively heterogeneous. Nonuniform slip causes an apparent change in the wavelength distribution of the initial free-surface elevation (i.e., the potential energy distribution), which affects the path of energy propagation. In the uniform slip scenario, the maximum free-surface elevation pattern is straightforward and clearly controlled by the topography. However, many strong and seemingly chaotic paths of wave energy appear in the nonuniform slip scenarios, and the free-surface field exhibits additional
uncertainties in terms of the flow. In Fig. 3b, the maximum free-surface elevation mainly propagates toward two places where the seafloor bathymetry becomes shallower relative to the deep areas northeast of Taiwan as shown in Fig. 2. Although the propagation paths due to the nonuniform slip distributions (Fig. 3d and 3f) also have the same characteristics, it is notable that the paths followed by the wave energy differ depending on the rupture pattern. To the northeast of Taiwan in Fig. 3f, there is a strong wave path connecting the two higher-elevation areas of bathymetry. However, this behavior is not observed in Fig.
3b and 3d. In addition, the maximum elevation on the footwall in Fig. 3d is higher than that in Fig. 3f. In Fig. 3b, the high elevation appears only along the coast on the footwall side. These results indicate that the wave energy variation depends on the rupture pattern, thereby causing differences in the wave paths and leading to completely different tsunami amplitudes.

### 3.2 Wave characteristics

Thirty stations located along the coastlines are available for recording the amplitude of the tsunami wave height. Relative to the other stations, stations 25 (Shihti), 26 (Hualien) and 27 (Suao) are situated near the potential rupture fault, and they have high wave amplitudes and enormous variations in the tsunami simulations of 100 different slip distributions; consequently, the time series of the wave heights at these stations are shown as an example (Fig. 4). The time series of the wave heights at the other stations are shown in the supplement. The variability in the distribution of the initial free-surface elevation results in
substantial phase changes and different wave heights. It is worth noting that the average of the disordered and chaotic time series produced by the 100 different slip distributions is almost identical to the results of the time series produced by the

uniform case. This implies that the uniform slip distribution simply represents an average result and that it cannot represent all of the possible situations.

According to the statistical results from 100 different slip patterns (Table 1) for 30 stations, Hualien station has the maximum wave amplitude of 7.32 m, and its maximum wave amplitude interval ranges from 1.87 to 7.32 m, which constitutes the widest interval for any recording site, and the standard deviation of this distribution is 1.024 m. These findings indicate that Hualien station has a high uncertainty in this scenario setting. However, the maximum wave amplitudes from the uniform slip distribution are relatively lower than those from the stochastic results. Following the above findings, we need to consider whether the estimations from the uniform slip case are appropriate for hazard analysis by focusing on the maximum wave amplitude issue.

### 3.3 The peak tsunami amplitude probability

According to the results of our simulations, we calculate the probability of the peak/maximum tsunami amplitude (PTA) at each recording station as shown in the histogram of Fig. 5. To verify the representativeness of the PTA probability distributions, another 100 sets of different slip distributions are produced and simulated under the same seismic conditions. In Fig. 5, the shapes of the PTA distributions from another 100 sets (black lines) are similar to the shapes of the histograms from the first 100 sets. These results verify the representativeness of the PTA probability distributions produced from 100 sets of slip distributions. This test also reinforces the reproducibility of our simulations and demonstrates that the number of simulations is roughly satisfactory for statistical analysis. Of course, the more slip distributions we use, the more comprehensive and stable the range we obtain.

In Fig. 5, the PTA distributions for the stations in eastern Taiwan (red markers) have obviously higher values than those in western Taiwan (blue markers) due to the specified location of the tsunami source. The shapes of the PTA distributions in eastern Taiwan resemble lognormal distributions, while those in western Taiwan resemble normal distributions. We suppose that the attenuation of the wave propagation causes the lognormal distributions to degenerate into normal distributions. The PTAs produced by a uniform slip distribution are generally located in the middle of the PTA distributions. Both PTA values (i.e., the value of the PTA from the uniform slip distribution and those from the stochastic slip distribution models) decrease with the distance from the potential fault due to attenuation of the wave propagation (Figure 5 shows the results for all stations, and Fig. 6 shows the results for stations 20 through 30 in eastern Taiwan). However, some stations, e.g., stations 17, 19, and 21, do not precisely follow this trend; this could be the result of the coastal topography and the presence of an energy channel. From Fig. 3d, in comparison with the adjacent coastline, station 21 is located exactly where the wave energy gathers. In addition, broad distributions are frequently observed at promontories along the coastline and are caused by complex propagation path effects between the source region and the recording locations (Geist, 2002). There are many compound

factors that affect the tsunami propagation and maximum wave height. Figure 6 presents the relation between the distance and wave height and shows the PTA distributions following Fig. 5. The x-axis presents the shortest distance between the stations and fault plane. On the footwall side, stations 20 and 22, which do not directly face the energy propagation path (Fig. 3f), are located on islands off the coast of Taiwan; consequently, their PTA distributions are lower than those of stations 21 and 23,
even though the distances from the potential fault are similar. On the hanging wall, station 29 is farther from the coastline of Taiwan than other stations; however, because of the real location of the station and its numerical grid setting, its PTA distribution is lower than that of station 30 (Fig. 3b). The ranges of the PTA distributions converge with increasing distance on the both sides of the fault. Moreover, the PTA distributions and their average values roughly exhibit a linear decrease with increasing distance except for stations 25 and 26. In contrast, these two stations in the near field are directly affected by initial
water elevation, and thus, the PTAs caused by uniform slip are quite low.

Although the seismic parameters have already been defined as constants in our experiment, there exists an uncertainty in the PTA, which is not a constant value. Hence, the uniform case cannot provide this uncertainty, and thus, the PTA could be underestimated. The results give specific PTA ranges, which represent the wave height uncertainties for the scenario of
earthquakes originating from the Ryukyu Trench. It is therefore necessary to consider the effects of a heterogeneous slip distribution to comprehensively assess the tsunami hazard.

## 4 Discussion

### 4.1 Tsunami

Most coastlines threatened by near field tsunamis, such as the coasts of Chile, Japan and Indonesia, are parallel to the trench axis of the associated subduction zones. Many tsunami events, including the Mw 8.8 Chile earthquake in 2010 (Lay et al., 2010; Fritz et al., 2011), the Mw 9.0 Tohoku earthquake in 2011 (Goda et al., 2015; Goda and Song, 2016), the Mw 9.1 Sumatra earthquake in 2004 (Lay et al., 2005), and the Mw 8.1 Mentawai earthquake in 2010 (Satake et al., 2013), have
occurred along these regions. However, the potential rupture fault in this study along the southernmost Ryukyu subduction zone is perpendicular to the coast of Taiwan which directly affects the first wave motion. The first motion on the footwall is up; conversely the first motion on the hanging wall is down. As a result, the coastline retreats from the land to the sea as the first tsunami wave approaches, allowing people additional time to leave the seafront.

The effect of a heterogeneous slip distribution is important and necessary to consider for near field estimations (Geist, 2002 and Ruiz et al., 2015). Figure 5 shows that the PTA distributions in the near field are broad, and they narrow with increasing

distance from the potential fault. The uncertainty in the near field is higher than that in the far field. At most of the eastern stations, the values of the average PTA approach uniform results, but the uniform slip results at stations 25 and 26 are close to the minimum PTA (Table 1). Geist (2002) presented the average and extreme nearshore PTA calculated for 100 different slip distributions and compared them with the uniform slip results (Fig. 6a in Geist (2002)). The range of the PTA also becomes

narrower with increasing distance. The values from the uniform slip distribution and the average PTA are similar, but some of the average values are close to the minimum PTA between approximately 19°N and 19.5°N. Similar characteristics of the average PTA and the results from the uniform case are observed in different regions. The average PTA is equal to the uniform slip result in the nearshore region, but this could be caused by other factors (e.g., distance to the tsunami source, propagation path, initial water elevation, etc.) that shift the average PTA toward the minimum PTA.

Four nuclear power plants (NPPs) are located on the island of Taiwan. According to the numerical results, we infer that the mean PTA in the coastal area of NPP4 ranges from approximately 2 to 3 m. The distribution at this plant may be wider than those at other nuclear power plants due to its position relative to the tsunami source. Moreover, NPP4 is located on the shore of a bay with a curved shape; the magnification effect from the geometrical shape of the bay may serve to enhance the PTA

therein. NPP3 also exhibits this condition insomuch that the energy is concentrated at the location of the plant (Fig. 3b, 3d and 3f). For the coastal areas around NPP1 and NPP2, the PTA distributions are between 1 and 2 m. The coastlines of these two nuclear power plants slightly face the direction of tsunami propagation, and thus, their PTAs should be higher than those along adjacent coastlines (Fig. 3b, 3d and 3f). In general, under this scenario, the coastline at NPP4 has the largest threat. Although NPP3 is far from the tsunami source, it faces a wave height of approximately 1.5 m on average with a ±0.5 m range of

uncertainty. However, NPP3 is closer to the Manila subduction zone, and thus, it could be threatened by a tsunami originating from the Manila Trench. In contrast, the coastlines of NPP1 and NPP2 are relatively safe and have fewer uncertainties with regard to the PTA.

The use of heterogeneous slip patterns clearly delineates the range of possible waveforms and provides more information on

latent uncertainties in the wave height. The 95% confidence intervals for the wave height from 100 sets in each time series provide us a specific range for the amplitude of the tsunami wave (Fig. 4). According to these time series, we are aware of the periods of tsunami runup and runoff and can prepare supporting policies to reduce associated disasters. For example, a nuclear power plant includes a trench from the ocean for the intake of water to cool the reactor; thus, if the sea level is too low to take in water, the temperature of the reactor will rise excessively, causing a nuclear disaster. Based on the results of simulations,

we can estimate that how much water should be stored for tsunami runoff. This issue requires more attention in Taiwan because four nuclear power plants are located near the coast.

## 4.2 Stochastic slip model

The results of the tsunami simulations illustrate that the effect of the slip distribution on the rupture plane has significant effects on the wave propagation and wave height. The correctness of this slip distribution determines whether the wave height calculations represent a useful reference. However, some parameters of the stochastic models could influence the synthetic slip distributions. For instance, the exponent of the slip spectrum is associated with the roughness of the slip distribution. Higher exponential values inhibit the powers of high wavenumbers, leading to smoother slip distributions; conversely, lower values lead to rougher slip distributions. In general, the $k$-square model needs to be followed. Furthermore, the interpolation of the slip distribution for a given geometry will affect the exponent of $k$ (Tsai, 1997). Interpolation will smooth the original pattern. The powers of short wavenumbers will be depressed and the powers of long wavenumbers will be enhanced. Moreover, the random spatial variability of the slip distribution is relatively critical. According to Lavallée and Archuleta (2003) and Lavallée et al. (2006), we adopt the truncated non-Gaussian distribution for the spatial variability. This truncation limits the non-Gaussian distribution to a particular range. However, extreme truncation will cause the heavy-tailed characteristic of this distribution to become less pronounced or even disappear, similar to a Gaussian distribution. In mathematics, the synthetic slip distribution is a filtering process insomuch that the characteristics of a heavy-tailed distribution affect the extrema of the slip distribution. The maximum slip will be greater as the truncated range increases, and the maximum slip may exceed reasonable values if the truncated range is excessively wide. Therefore, the parameters must be chosen carefully to match the observations acquired by inversion.

## 5 Conclusion

The maximum possible earthquake magnitude is $M_w$ 8.15 with an average slip of 8.25 m in the southernmost portion of the Ryukyu Trench. One hundred slip distributions of the seismic rupture surface were generated by a stochastic slip model. The maximum slip range is between 20.17 and 37.97 m, and the average slip of each model is consistent with 8.25 m. A heterogeneous slip distribution induces variability in the tsunami wave heights and the associated paths of propagation. The simulated results demonstrate that the complexity of the rupture plane has a significant influence on the near field for local tsunamis. The PTA distribution provides a specific range for the wave height and the probability of occurrence in this scenario. These distributions and their average values exhibit an approximately linear decrease with increasing distance. The coastline, which is situated very close to or even atop the tsunami source, is directly affected by the rupture slip distribution. Then, the range of the PTA distribution will converge with increasing distance from the tsunami source. In this study, Hualien station, which is located directly above the source, has the widest PTA interval (1.87-7.32 m) and the highest wave amplitude. The statistical summary reveals that this station, whose standard deviation is 1.63 m, which is larger than those of the other stations, has the largest uncertainty. However, the PTA caused by the uniform slip distribution is only 1.63 m, which is much lower and is even below the average (3.36 m) at this station. This finding implies that a simplified earthquake source cannot completely

represent the tsunami amplitudes in reality. If we adopt a uniform slip distribution to assess tsunami hazards, those hazards will be critically underestimated. Furthermore, the variances of tsunami amplitudes are imperative for assessing tsunami hazards, and the quantitative technique employed is also important.

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

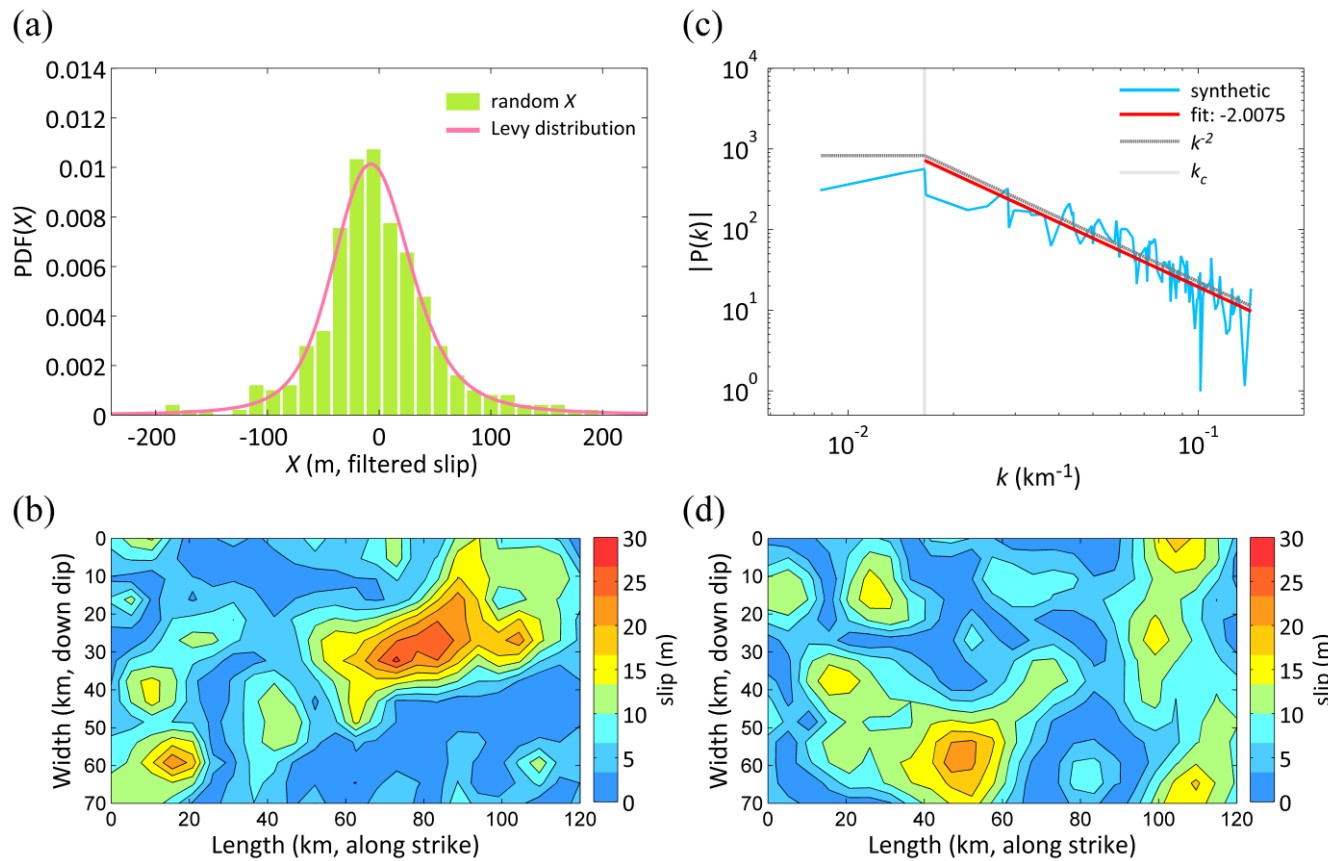

**Figure 1. (a)** The spatially random variable: truncated Lèvy distribution. The Lèvy parameters obtained from the Northridge earthquake were taken from Lavallée et al (2006). **(b)** A stochastic slip distribution is generated by filtering the spatial random variable $X$ in Fig. 1a. This slip pattern produces the highest maximum wave amplitude at Hualien station. **(c)** The slip spectrum is calculated from Fig. 1b. This slip spectrum decays with an exponent of -2 according to a characteristic corner radial wavenumber. This verifies that the synthetic slip distribution is identical to the $k$-square model and the condition of the rupture dimension. **(d)** This stochastic slip distribution produces the lowest maximum wave amplitude at Hualien station.

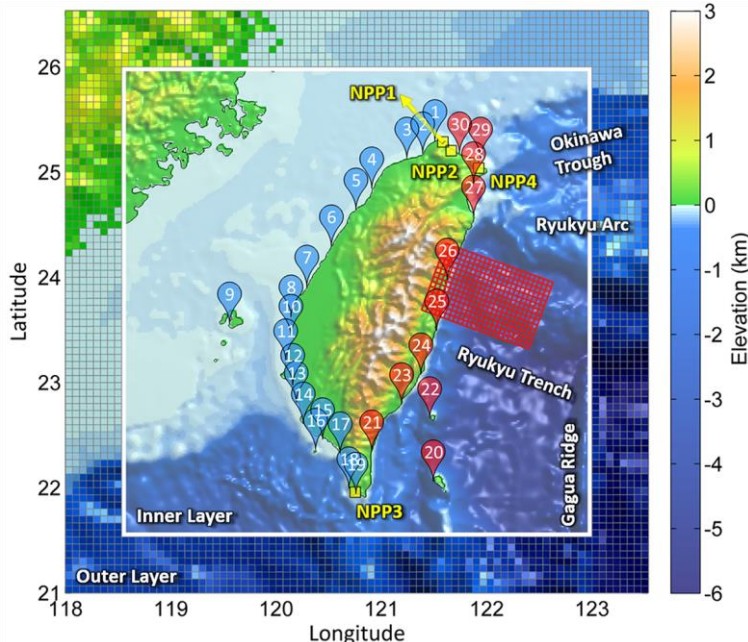

**Figure 2. The map of Taiwan shows the fault model and recording stations used in this study. The bathymetry is divided into 2 layers with different resolutions. The resolution of the outer layer is 4 minutes, and the resolution of the inner layer is 1 minute. The red grid denotes the potential fault model (5×5 km² grid size). The pins represent 30 tidal gauges of the CWB. The red and blue colors indicate stations on the eastern and western sides of Taiwan, respectively, and the yellow squares represent the sites of nuclear power plants.**

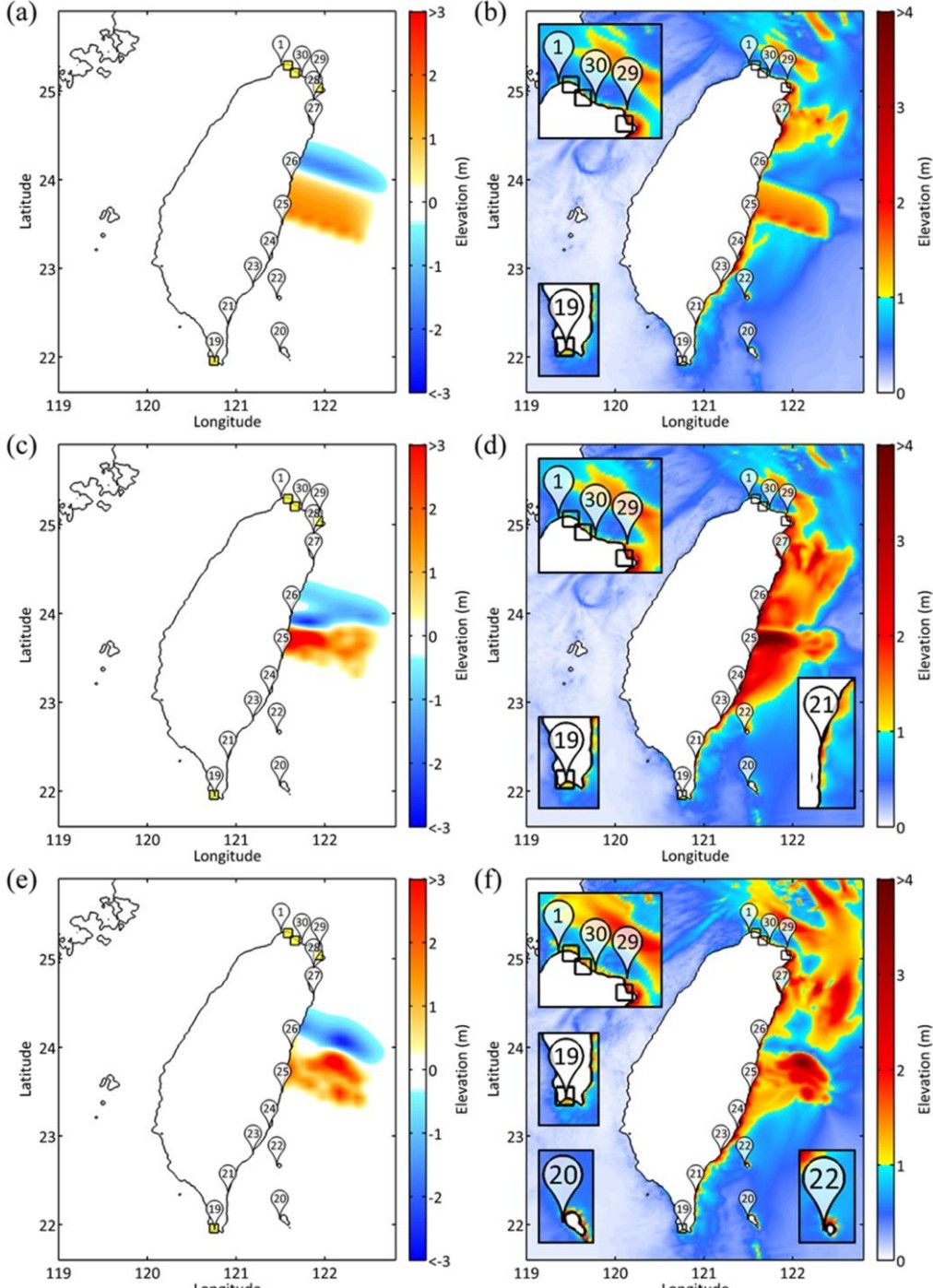

**Figure 3. (a), (c) and (e) are the initial water elevations, and the color bars represent the elevation of the initial water surface. (b), (d) and (f) are the maximum free-surface elevation (i.e., the distribution of the energy path), and the color bars represent the elevation of the maximum free-surface. (a) and (b) displays the results with a uniform slip distribution. (c) and (d) displays the results from Fig. 1b. (e) and (f) displays the results from Fig. 1d. The seafloor elevation fundamentally dominates the tsunami propagation, but the**

slip distribution also has a strong influence. In (a, c and e), yellow squares represent NPPs; in (b, d and f), the NPPs are represented by open squares.

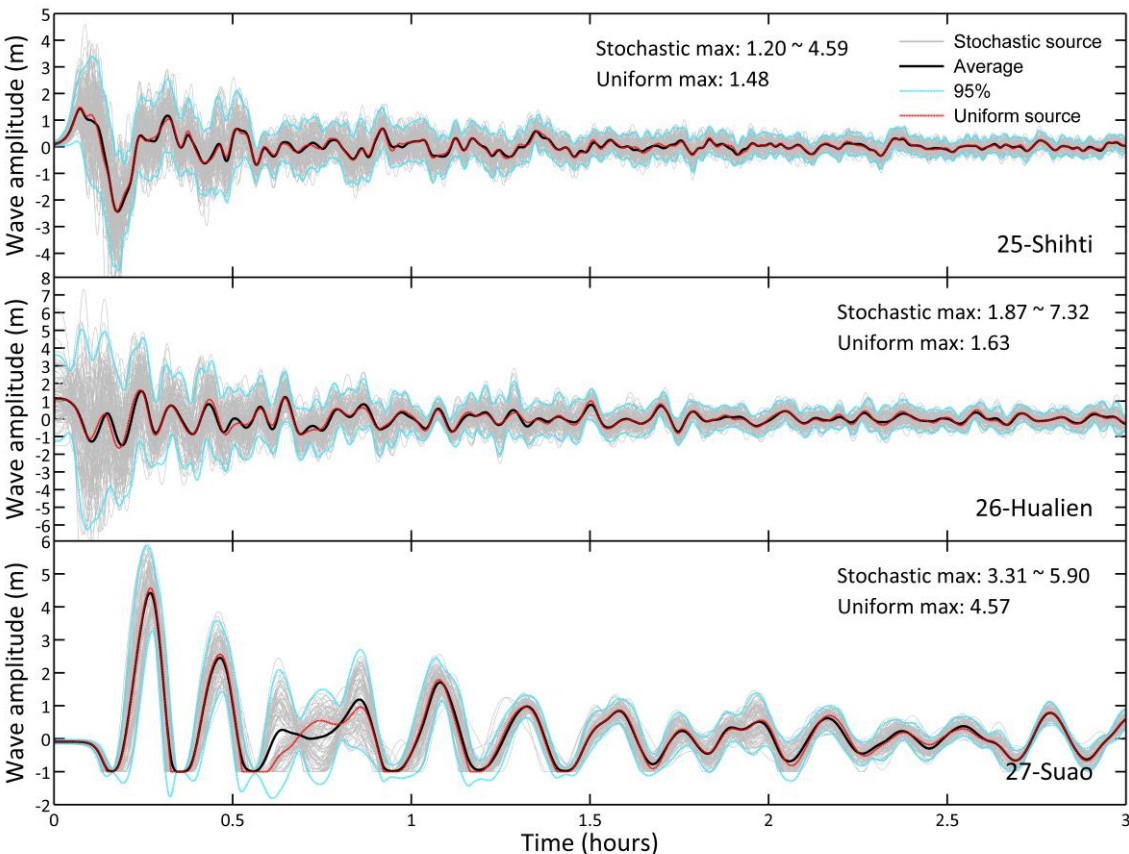

5 **Figure 4. The time series of the wave heights recorded at stations 25 (Shihti), 26 (Hualien) and 27 (Suao). Gray lines represent the time series of 100 different slip distributions; black lines represent the averages of the gray lines; blue lines represent the 95% confidence intervals; and red lines are the time series produced using uniform slip distribution. Parts of the wave heights at station 27 are lower than the water depths, and thus, these curves have been truncated.**

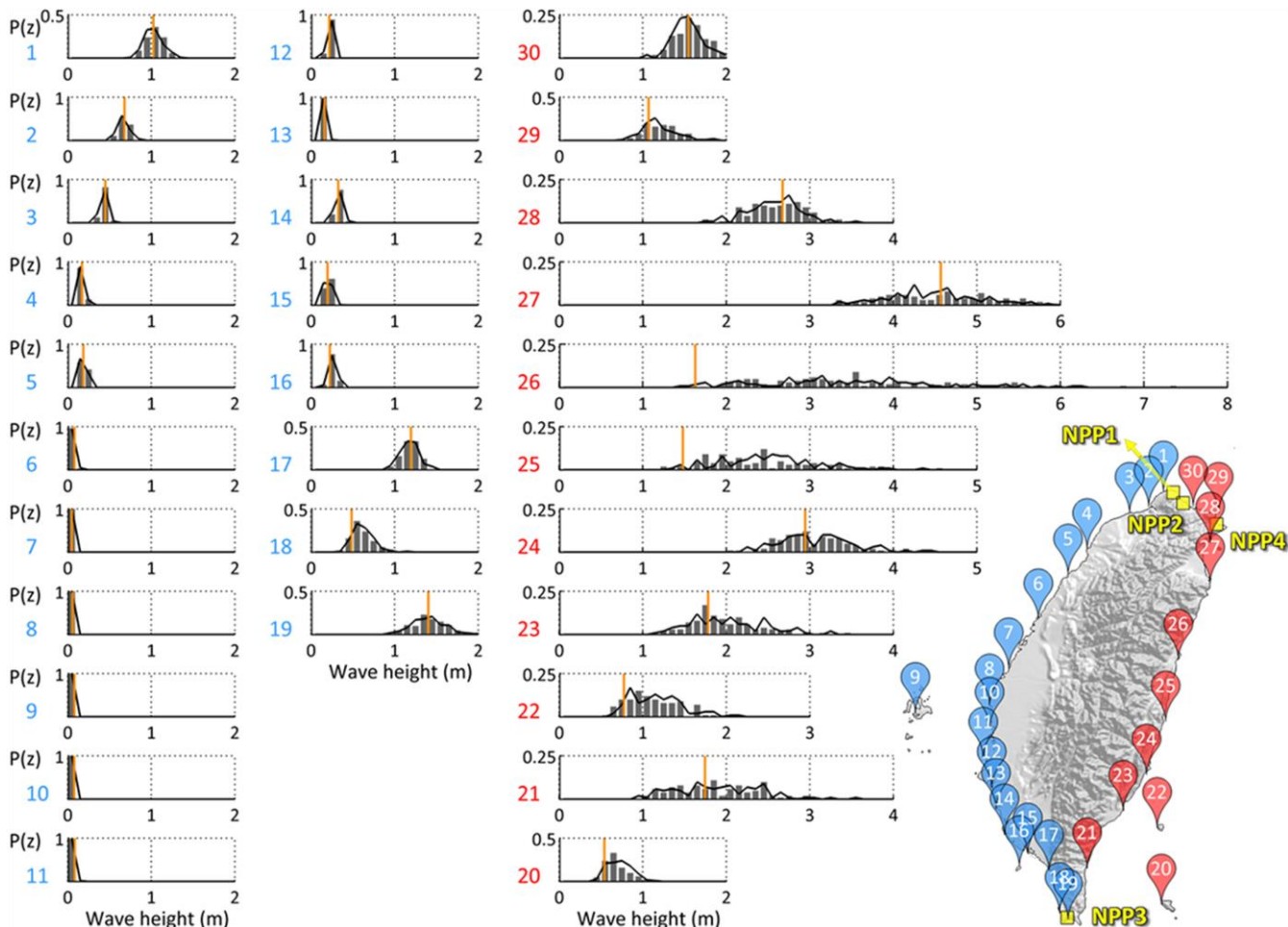

**Figure 5. The probabilities of the PTA along the coast of Taiwan (blue: stations 1~19, red: stations 20~30). The histograms display the PTAs derived from 100 different slip simulations. The black lines represent the results from another 100 simulations, and the orange lines represent the PTA obtained using a uniform slip distribution. The PTA probability distribution gives a clear PTA range and its occurring probability. The map of Taiwan shows the station locations and the sites of four NPPs (yellow squares).**

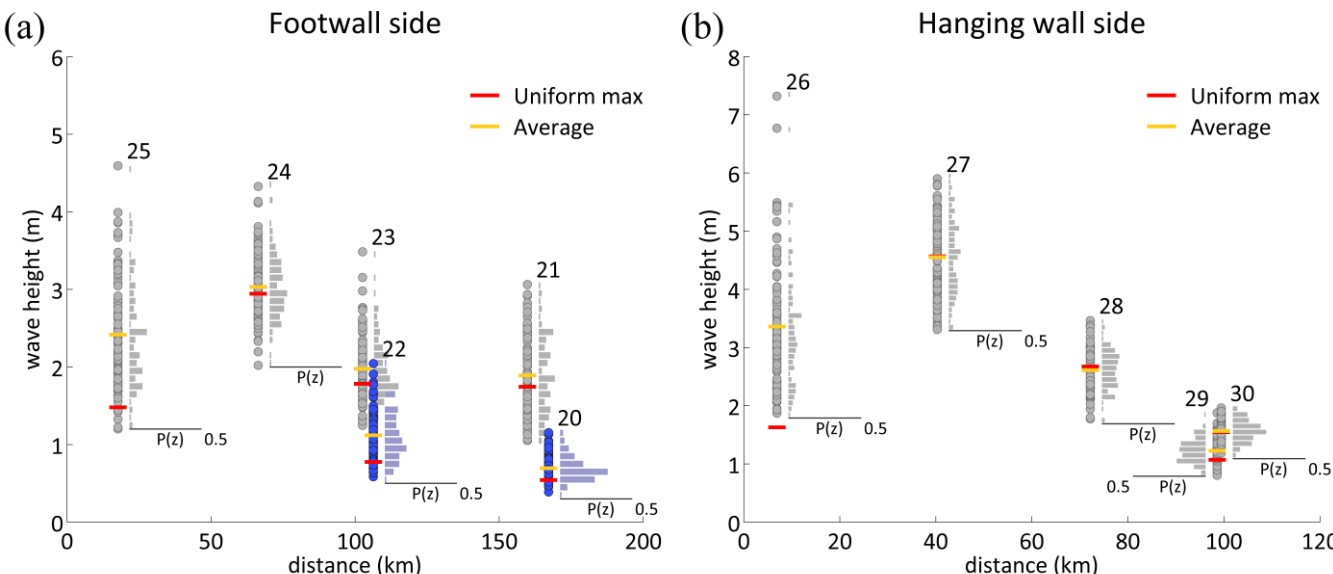

**Figure 6. The relation between the distance and wave height for stations 20 through 30 in eastern Taiwan. (a) is the station on the footwall side. Stations 20 and 22 (blue color) are off the shoreline of Taiwan. (b) represents the stations on the hanging wall side. Both sides roughly exhibit a linear decay and range of uncertainty converging with increasing distance for the tsunami amplitude. Red bars show the PTA of the uniform slip distribution, and yellow bars show the average of the PTAs from the stochastic slip models.**

**Table 1. The maximum, minimum, and average wave heights with their standard deviations for the PTA probability distributions (in meters) with the maximum wave heights from the uniform slip model. The water depths at the stations in the computational mesh are also included.**

| # | Station | Lon. | Lat. | Min [m] | Max [m] | σ [m] | Avg. [m] | Max [m] (uniform slip) | Water depth [m] |
|---|---------|------|------|---------|---------|-------|----------|------------------------|-----------------|
| 1 | Linshanbi | 121.5106 | 25.2844 | 0.80 | 1.32 | 0.108 | 1.04 | 1.02 | 4.00 |
| 2 | Danshuei | 121.4019 | 25.1844 | 0.55 | 0.83 | 0.061 | 0.68 | 0.68 | 4.00 |
| 3 | Jhuwei | 121.2353 | 25.1200 | 0.33 | 0.52 | 0.039 | 0.44 | 0.45 | 1.75 |
| 4 | Hsinchu | 120.9122 | 24.8503 | 0.13 | 0.24 | 0.025 | 0.17 | 0.17 | 3.50 |
| 5 | Waipu | 120.7717 | 24.6514 | 0.15 | 0.26 | 0.020 | 0.20 | 0.19 | 0.50 |
| 6 | Taichung Port | 120.5250 | 24.2917 | 0.07 | 0.11 | 0.009 | 0.08 | 0.08 | 0.00 |
| 7 | Fanyuan | 120.2972 | 23.9147 | 0.04 | 0.06 | 0.004 | 0.05 | 0.05 | 1.00 |
| 8 | Bozihliao | 120.1417 | 23.6250 | 0.05 | 0.07 | 0.004 | 0.06 | 0.06 | 0.00 |
| 9 | Penghu | 119.5669 | 23.5636 | 0.07 | 0.09 | 0.005 | 0.08 | 0.08 | 1.00 |
| 10 | Dongshih | 120.1417 | 23.4417 | 0.06 | 0.09 | 0.005 | 0.08 | 0.08 | 1.00 |

| 11 | Jiangjyun | 120.1000 | 23.2181 | 0.06 | 0.10 | 0.007 | 0.09 | 0.09 | 0.00 |
|----|-----------|----------|---------|------|------|-------|------|------|------|
| 12 | Anping | 120.1583 | 22.9750 | 0.15 | 0.26 | 0.018 | 0.22 | 0.22 | 0.00 |
| 13 | Yongan | 120.1917 | 22.8083 | 0.11 | 0.20 | 0.016 | 0.16 | 0.16 | 5.25 |
| 14 | Kaohsiung | 120.2883 | 22.6144 | 0.23 | 0.43 | 0.039 | 0.33 | 0.33 | 2.00 |
| 15 | Donggang | 120.4417 | 22.4583 | 0.15 | 0.28 | 0.026 | 0.21 | 0.20 | 7.25 |
| 16 | Siaoliouciou | 120.3750 | 22.3583 | 0.17 | 0.40 | 0.046 | 0.26 | 0.22 | 12.75 |
| 17 | Jiahe | 120.6083 | 22.3250 | 0.90 | 1.44 | 0.098 | 1.19 | 1.20 | 4.00 |
| 18 | Syunguangzuei | 120.6917 | 21.9917 | 0.33 | 0.96 | 0.124 | 0.61 | 0.49 | 64.25 |
| 19 | Houbihu | 120.7583 | 21.9417 | 0.90 | 1.96 | 0.197 | 1.41 | 1.40 | 15.50 |
| 20 | Lanyu | 121.4917 | 22.0583 | 0.39 | 1.15 | 0.155 | 0.69 | 0.54 | 347.50 |
| 21 | Dawu | 120.8972 | 22.3375 | 1.05 | 3.06 | 0.487 | 1.89 | 1.74 | 31.75 |
| 22 | Lyudao | 121.4647 | 22.6622 | 0.58 | 2.04 | 0.316 | 1.12 | 0.78 | 146.00 |
| 23 | Fugang | 121.1917 | 22.7917 | 1.25 | 3.48 | 0.409 | 1.98 | 1.78 | 24.00 |
| 24 | Chenggong | 121.3767 | 23.0889 | 2.02 | 4.33 | 0.416 | 3.03 | 2.94 | 32.50 |
| 25 | Shihti | 121.5250 | 23.4917 | 1.20 | 4.59 | 0.680 | 2.42 | 1.48 | 142.75 |
| 26 | Hualien | 121.6231 | 23.9803 | 1.87 | 7.32 | 1.024 | 3.36 | 1.63 | 37.00 |
| 27 | Suao | 121.8686 | 24.5856 | 3.31 | 5.90 | 0.641 | 4.55 | 4.57 | 1.00 |
| 28 | Gengfang | 121.8619 | 24.9072 | 1.78 | 3.47 | 0.337 | 2.61 | 2.67 | 24.00 |
| 29 | Longdong | 121.9417 | 25.1250 | 0.80 | 1.88 | 0.202 | 1.23 | 1.07 | 60.75 |
| 30 | Keelung | 121.7417 | 25.1750 | 1.19 | 1.96 | 0.183 | 1.57 | 1.55 | 15.50 |