# Peer review of "Assessment of the peak tsunami amplitude associated with a great earthquake occurring along the southernmost Ryukyu subduction zone in the region of Taiwan"

_Natural Hazards and Earth System Sciences, 2017_

## Referee Comment (RC1) · Anonymous Referee #1 · 24 Nov 2017

First of all, I want to point out that I can only review the tsunami modeling part of the preprint. I cannot comment on the seismological part, the choice of stochastic sources.
* * *
*1. Does the paper address relevant scientific and/or technical questions within the scope of NHESS?*

Yes.

[Figure]

*2. Does the paper present new data and/or novel concepts, ideas, tools, methods or results?*

To my knowledge, it is the first published assessment of tsunami amplitudes of this kind for Taiwan. The setting that the coast line is perpendicular to the rupture is interesting in general.
* * *
*3. Are these up to international standards?*
*4. Are the scientific methods and assumptions valid and outlined clearly?*
*7. Is the description of the data used, the methods used, the experiments and calculations made, and the results obtained sufficiently complete and accurate to allow their reproduction by fellow scientists (traceability of results)?*

Concerning the application of the tsunami model: No. COMCOT is used as a black box. My major criticism is that the model is not validated for this area, and I strongly suggest to add a hind cast of a real event to prove that COMCOT with the chosen settings delivers realistic simulations. Probably, the last near field tsunami in 1867 is not well suited for a hind cast due to the lack of measurements, but the Tōhoku tsunami 2011 should be a good test case also for Taiwan.

The following questions should be addressed:

- Which formulas and parameters are used, in particular for bottom friction (Manning coefficient)? The bottom friction has an impact on the simulated tsunami amplitude at the coast.

- Which bathymetry and topography data is used? Free GEBCO and SRTM?

- The resolution of 1 minute for the inner mesh is quite rough for simulations that should give estimates of the tsunami amplitude at the coast. Our experience

from hind casts of real events suggests that at the coast line, the horizontal resolution should be 500m (edge length in an unstructured triangular grid) or better. This should be transferable, as COMCOT also is a model with first order spatial discretization.

- Where are the tide gauges located? See also point 14, references. On the one hand, the exact location is not really important, because the study could be performed with virtual sensor locations or coastal forecast points, but

  – to reproduce the results, the locations of the (real or virtual) gauges are needed,
  – for hind casts of real events, the location and measurements from real tide gauges are needed,
  – the simulation of the tsunami wave form at a tide gauge that is located e.g., inside a harbor or narrow bight is very sensitive to errors in the representation of bathymetry and topography (1min resolution for sure is too coarse!) and to the choice of the roughness parameter (wave reflections).
  – The comparison in fig. 6 may be spoiled by different gauge locations. Distance to the source is not the only parameter, as it is also stated in the paper, too (e.g., page 7 line 23-24).
* * *
*5. Are the results sufficient to support the interpretations and the conclusions?*
*6. Does the author reach substantial conclusions?*

Yes, but keep in mind that the results were obtained by a black box simulation.
* * *
*8. Does the title clearly and unambiguously reflect the contents of the paper?*

*9. Does the abstract provide a concise, complete and unambiguous summary of the work done and the results obtained?*
*10. Are the title and the abstract pertinent, and easy to understand to a wide and diversified audience?*

Yes.
* * *
*11. Are mathematical formulae, symbols, abbreviations and units correctly defined and used? If the formulae, symbols or abbreviations are numerous, are there tables or appendixes listing them?*

Equation (1): W for width, L for length: It's obvious, but nevertheless should be added in the text above.

Which value for $\mu$ is assumed when estimating Mw? And as a non-seismlogist, I would like to ask if the estimate of $\bar{D}$ = 8.25m is really obvious?

Section 2.2: Not my field of expertise at all.
* * *
*12. Is the size, quality and readability of each figure adequate to the type and quantity of data presented?*

Figure 4: change y-axis label to "Wave amplitude"

Figure 6: I would keep this figure, but skip the explicit linear fitting. It pretends an accuracy that cannot be obtained.
* * *
*13. Does the author give proper credit to previous and/or related work, and does he/she indicate clearly his/her own contribution?*

To me, the distinction between own work and work from other scientists seems clear.

*14. Are the number and quality of the references appropriate?*

A citation for the tide gauge locations or at least a list of coordinates would be handy. The Taiwanese tide gauges are not available at
`http://www.ioc-sealevelmonitoring.org` or `http://www.psmsl.org/` (Taipei until 1995, Kaohsiung until 1996), and I could not find a link to the gauges at the website of the Taiwanese Central Weather Bureau (CWB) `http://www.cwb.gov.tw` This private/commercial site was the best information I could find:
`https://www.tide-forecast.com/locations/Hualien-City`. Still, no exact location, but the "Detailed Map" gives at least an idea that this station is located inside the harbour. In total, 9 Taiwanese stations are available here.

I am missing a short overview of historical tsunamis in Taiwan, but the last local tsunami occurred in 1867, and it might be difficult to find scientific papers to cite, see e.g.,
`http://scweb.cwb.gov.tw/NewsContent.aspx?ItemId=37&CId=199&loc=en`

However, I found the following paper - no tsunami, but a report on the uplift of the tide gauge due to the earthquake. Maybe, this paper provides a helpful hindcast, too: COMCOT should not show a strong tsunami.

Chung-Liang Lo, Emmy Tsui-Yu Chang, and Benjamin Fong Chao. Relocating the historical 1951 Hualien earthquake in eastern Taiwan based on tide gauge record. Geophys. J. Int. (2013) 192, 854–860. doi: 10.1093/gji/ggs058

*15. Are the references accessible by fellow scientists?*

Yes, but please add doi numbers.

[Figure]

*16. Is the overall presentation well structured, clear and easy to understand by a wide and general audience?*
*17. Is the length of the paper adequate, too long or too short?*

Yes, and adequate except that more explanation is needed for the tsunami simulation.
* * *
*18. Is there any part of the paper (title, abstract, main text, formulae, symbols, figures and their captions, tables, list of references, appendixes) that needs to be clarified, reduced, added, combined, or eliminated?*

Figures: See remark for point 12.
* * *
*19. Is the technical language precise and understandable by fellow scientists?*
*20. Is the English language of good quality, fluent, simple and easy to read and understand by a wide and diversified audience?*

As a non-native speaker, it is not easy to comment on the language. I observed several grammatical errors, but overall, I found the article easy to read and understand. In particular in the tsunami context, the scientific language is precise.
* * *
*21. Is the amount and quality of supplementary material (if any) appropriate?*

N/A
* * *

---

## Referee Comment (RC2) · Anonymous Referee #2 · 10 Dec 2017

Assessment of peak tsunami amplitude associated with a great earthquake occurring along the southernmost Ryukyu subduction zone for Taiwan region

Yu-Sheng Sun, Po-Fei Chen, Chien-Chih Chen, Ya-Ting Lee, Kuo-Fong Ma and Tso-Ren Wu

Nat. Hazards Earth Syst. Sci

=============== General comments ===============

The manuscript presents an interesting application of tsunami modeling in the subduction zone of Taiwan, where the trench is orthogonal to the coastline of Taiwan island. Authors assume a potential earthquake of magnitude Mw 8.15 breaking a fault of dimension 120 x 70 km2 with an average slip of 8.25 m. Several tsunami scenarios are simulated using slip models generated using stochastic random fields where the power spectrum of slip is assumed to falloff as k-2 at high wavenumber. The authors focus their analysis in the variability of the maximum tsunami wave height computed at several sites located around the Taiwan island. The histograms computed at each sites show a great variability, in particular at the eastern border of Taiwan, and these results are also compared against the maximum tsunami wave height computed assuming an uniform slip rupture scenario. Results reveal that assuming an uniform slip the maximum tsunami wave height underestimates a large number of predictions from nonuniform slip scenarios, but not at all sites. Also, the tsunami wave propagation shows a complex pattern of energy radiation, mostly due to bathymetry, but also to slip distribution. The subject is of general interest and with interesting results obtained in the near field, where the whole deformation pattern of an inverse fault affects the static displacement field of the coastal border along the eastern border of Taiwan island. There are some inconsistencies in the random slip generation that must be clarified, the choice of the parameters needs some justification, and few aspects that must be clarified and discussed. Then, from my point of view it requires important revisions before publication.

I apologize because I am not native English speaker, so I can not help much on it, but from my own expertise, I feel that the authors should carefully check the grammar, spellchecking, some figures and text, which are in some parts could be better executed.

=================== Major and minor points ===================

Page 2, lines 3-5. If the earthquakes associated to the historic tsunamis mentioned in the text have any magnitude estimation, please provide the value and include the reference. For instance, the 1867 tsunami, magnitude ?.

Page 2, lines 13-15. When comparing PTHA and PSHA, authors mentioned in the text that PSHA works with ground-motion parameters. So, can you complete the idea by specifying that PTHA works with tsunami wave amplitudes, or some other wave measurements?. If there is any reference, please include it.

Page 3. Line 2. Please, provide the reference for the magnitude range, Mw 7.5-8.7.

Page 3. Line 12. About the fault geometry setting. Which is the source depth of the top (or bottom) of the fault plane ?. I think it has not been specified yet in the text.

Page 3. Line 15, please complete to "...in dip slip faults".

Page 3. Eq. (1), please, specify what is L, and W.

Page 3. Line 18. I suggest to change "constant" by "parameter". Strictly speaking, in elastic heterogeneous media, the Lamè parameters (lambda and mu) vary in space.

Page 3. In Eq. (2). Which is the value assumed for mu ?.

Page 3. Section 2.1. When the authors compute the earthquake magnitude, average slip and fault area. Did the authors compare (or contrast) these values with any magnitude/fault-size scaling relationship for subduction earthquakes ?. It could be interesting to compare these values with any magnitude/size scaling relationship for subduction zones.

Page 3, line 25. For completeness purposes, please provide the scalar seismic moment, M0 for the corresponding Mw 8.15.

Page 4. Please clarify or complete the sentence in line 8, because there is a dot at the end of the sentence, so it is not clear what Eq. (4) means or represents. The 2D Fourier spectrum amplitude of what ?.

Page 4. Line 10. Please, to be consistent with the notation in Eq. (4), please clarify the meaning of "F", or, change F by Fs,t which represents the 2D discrete Fourier transform of Dx,y. Also, for completeness purposes, specify that Dx,y is the slip distribution over

a 2D lattice, for instance.

Page 4. In line 10, please complete, "...wave number.", by "...radial wavenumber.".

Page 4. Line 13, please correct "corner frequency" by "corner radial wavenumber", because kc is not a frequency.

Page 4. Line 14. What happen with the phase beyond kc ?. Please, clarify. Or, the last sentence "Within the kc,....(Geist, 2002)." could be deleted because authors are describing the overall characteristics of the slip and not describing the details of how the random slip is generated numerically in the practice.

Page 4. Eq. (5). Please, be careful and clear with the mathematical notation. What does Fˆ(-1) represent ?. Is it the inverse 2D discrete Fourier transform ?.

Page 4. Line 23. Please, specify that PDF is Probability Density Function, I think it has not been mentioned before in the text.

Page 5. Line 3. Complete the units in the sentence, "...5x5 km...", by "...5x5 kmˆ2...".

Page 5. Line 3. Please, clarify that 24x14 are along strike and dip respectively.

Page 5. Line 1-4. I will ask the authors to provide some details about how the stochastic slip distribution is generated, and to be clear on the choice of parameters and discuss about the results. Please, read the following comments.

The authors used the values of the Levy PDF suggested by Lavallee et al. (2006), so please clarify in the manuscript that those values were estimated from an stochastic 2D model in the dip slip direction, obtained for the Northridge earthquake. So, why do you use parameters from a shallow crustal earthquake occurred in California to characterize a interplate subduction zone earthquake ?. Please justify, or discuss.

Notice that according to Lavallee et al (2006) and others, the scaling exponent is (nu+1) so, the Power Spectrum Density of slip is, P(k) ~ kˆ(-(nu+1)), it implies that the slip spectrum behaves as, D(k) ~ kˆ(-(nu+1)/2).

The authors generate random variables using the Levy distribution, and imposed $P(k) \sim k^{-2}$ as shown in Fig. 1c, so, the slip in the wavenumber domain behaves as, $D(k) \sim k^{-1}$, and Figure 1 is ok, but the slip spectrum does not follow the $k^{-2}$ source characteristic discussed at the beginning of Section 2.2. Please, clarify this point in the text. Also, discuss the effect in the spatial distribution of slip of this choice (falloff as $k^{-1}$) of the slip spectrum amplitude in the wavenumber domain), versus a slip spectrum that falloff as $k^{-2}$.

From the results shown in Fig. 1, authors generated a slip spectrum that decays as $k^{-1}$ because they imposed the power spectrum density as $P(k) \sim k^{-2}$, but in the legend they say "This slip spectrum decays with exponent of -2 and...", so, it is an inconsistency for me. Please, be clear on the choice, and the terminology used when generating spatial random fields. Herrero & Bernard (1994), Andrews (1981), and others, used a stochastic slip model with a 2D Fourier spectrum that decays as $k^{-2}$ which means, $D(k) \sim k^{-2}$. I am not saying the authors are wrong in their choice, it is only that some parts of the text need some clarification, justification of the choice, or discussion about the assumptions done.

Page 5. Line 3. Why did you set a 5x5 subfault size ?. Did you test different subfault sizes ?.

Page 5. Line 3. Did you assume a constant slip at each subfault ?. If it is the case, how do you treat the non-smooth slip boundary condition at the boundaries of the fault ?. Did you apply a taper at all the borders, if not, authors should discuss or justify their treatment.

Page 5, line 15. I would suggest to use "computational domain" instead of "...numerical model".

Page 5. Line 15. Complete the units in 5x5 km$^2$.

Page 5. Lines 15-19. Same comment as done in Page 5, line 3, about the assumption

of uniform slip at each subfault.

Page 5, lines 21-25. Why do you use 4 min and 1 min for the nested grids ?. Did you test a different grid size?. Which bathymetry/topography is used in the numerical simulation of the tsunami ?. Please include a reference. For instance, GEBCO (https://www.gebco.net/) provides a global 30 arc-sec bathymetry, which has a better resolution than the bathymetry used in this work. Please comment on it. Which is the boundary condition set at the coastlines (the boundary between wet and dry domains) ?. Do you assume a vertical wall condition, or do you allow inundation ?. Did you impose any friction, if yes, which one is the Manning's coefficient used in the simulation ?.

Page 6. Sentence in line 5-6 is a bit confusing, please rephrase to clarify.

Page 6. Section 3.1. If I understand, authors used the vertical seafloor displacement as initial condition to propagate the tsunami, and the horizontal motion of the seabed is not included in the simulation. I will suggest to clarify better these assumptions in Section 3.1.

Page 7. Section 3.3. Authors say basically that they computed the probability of the PTA by histograms, but from my understanding they show (Fig. 5) a probability density estimated from the numerical PTA data. I think authors could say/argue a little bit more about this, in terms of this choice and analysis. I mean, does the data follow any distribution (e.g. Gaussian, Levy, Log-normal) ?, Are the PTA data (simulated) Gaussian distributed ?. Is it possible to estimate the probability of exceeding a certain input value from these numerical results ?. I think some of these aspect are not discussed or mentioned in the text.

Page 7. Line 11. Please complete the idea that after generating the second set of slip models, the tsunami is simulated.

Page 7. Paragraph 3. When you compare PTA versus distance, how do you define or

measure the distance between source and station ?. At least, it could be mentioned or discussed in the text.

Page 8. Lines 14-16. Please, provide the references for the Maule, Tohoku and Sumatra earthquakes.

Page 8. Lines 22-31. The results discussed here are obtained at several sites, but It is not clear where the sites (tides gauges) are exactly located, right at the boundary, or surrounded by a wet domain even during the tsunami evolution ?. If the latter is true, the comparison of maximum tsunami wave height (this study) is not exactly straightforward comparable to runup (analyzed in other studies). Also, authors should comment on the effect (or limitations) of the grid resolution (1 arc-min, used in this study) over the results obtained. I suspect this coarse grid may have an effect on the simulations near the coast.

Page 8, line 17. Clarify what "lecture" means.

Page 9. Line 29. I would suggest to complete the idea in the sentence, "Furthermore, interpolation has a tremendous effect for the exponent value becoming larger with grid size reducing (Tsai, 1997).", because it refers to how the exponent and correlation lengths are computed from the solutions of slip models of earthquakes. On the other hand, some authors assume k-2 slip models based on other physical considerations.

============== Figures ==============

Figure 1. Clarify units, X ?, k ? , length km or 5km ?. To avoid misunderstanding, I suggest to delete the label "Northrigde earthquake" in the Fig 1a, and you can mention it in the caption (e.g. Levy parameters were taken from Lavallee et al......obtained for the Northridge earthquake.), because the realization shown is for an Mw 8.15 earthquake and not for the Northridge earthquake. Fault axis along dip and strike are confusing too. I will suggest to plot the real distance along strike and dip directions (with the correct units) and not the "indexes" of each subfault. What do represent the colorbar ?.

See my comments about P(k) and D(k), what is shown in Fig 1c is not what is written in the caption.

Figure 2. I suggest to contextualize at the beginning the region of the study area, (e.g. Map of Taiwan...for example). Correct 5x5 km by 5x5 km^2. Is the white box the nested inner grid ?. Colorbar ?.

Figure 3. I would suggest specify that the "energy propagation" corresponds to, maximum tsunami wave height, for instance. Colorbar ?.

======= Tables =======

Table 1. The description of the table and caption is a bit confusing. What is the meaning of Max(uni) ?. A suggestion is that a part of the description given at the end of the table can be moved to the caption, and authors can put the units [m] directly beneath each variable description.

---

## Author Comment (AC1) · 12 Feb 2018

We would like to thank Reviewer for the detailed revision and important suggestions. We improved the manuscript following suggestions. The attachment is modified manuscript.

Q:

3. Are these up to international standards?

4. Are the scientific methods and assumptions valid and outlined clearly?

7. Is the description of the data used, the methods used, the experiments and calculations made, and the results obtained sufficiently complete and accurate to allow their reproduction by fellow scientists (traceability of results)?

Concerning the application of the tsunami model: No. COMCOT is used as a black box. My major criticism is that the model is not validated for this area, and I strongly suggest to add a hind cast of a real event to prove that COMCOT with the chosen settings delivers realistic simulations. Probably, the last near field tsunami in 1867 is not well suited for a hind cast due to the lack of measurements, but the Tohoku tsunami 2011 should be a good test case also for Taiwan.

A: We have added some references in text. [Page 6, lines 2-6]

To solve the time dependent tsunami propagation, we adopt a well-validated numerical model, COMCOT (Cornell Multi-grid Coupled Tsunami Model). COMCOT is able to solve both linear and nonlinear shallow water equations on a Cartesian or Spherical coordinate systems (Wang 2009). In terms of validation, COMCOT has been widely used in studying many historical tsunami events, such as 1960 Chilean tsunami (Liu et al., 1995), 1992 Flores Islands tsunami (Liu et al., 1995), 2003 Algeria tsunami (Wang and Liu, 2005), 2004 Indian Ocean tsunami (Wang and Liu, 2006, 2007), and 2006 Ping-Tung tsunami (Wu, et al., 2008; Chen, et al., 2008). Taking the explicit leap-frog scheme to solve shallow water equation, COMCOT has the 2nd order accuracy in both special and time domains. COMCOT also supports the nested grid system that the finer grid can be placed on a coarser grid to increase the resolution locally. Thus, we can use finer grid in near-shore region and coarser grid in deep sea region.

Reference:

Chen, P. F., Newman, A. V., Wu, T. R., and Lin, C. C. (2008). Earthquake Probabilities and Energy Characteristics of Seismicity Offshore Southwest Taiwan. Terr. Atmos.

Ocean. Sci., 6, 697-703, doi: 10.3319/TAO.2008.19.6.697(PT)

Liu, P. L. F., Cho, Y. S., Yoon, S. B., and Seo, S. N. (1995). Numerical simulations of the 1960 Chilean tsunami propagation and inundation at Hilo, Hawaii. In Tsunami: Progress in prediction, disaster prevention and warning (pp. 99-115). Springer, Dordrecht. https://doi.org/10.1007/978-94-015-8565-1_7

Liu, P. L. F., Cho, Y. S., Briggs, M. J., Kanoglu, U., and Synolakis, C. E. (1995). Runup of solitary waves on a circular island. J. Fluid Mech., 302, 259-285. doi: 10.1017/S0022112095004095

Wang, X. (2009). User manual for COMCOT version 1.7 (first draft). Cornel University, 65.

Wang, X., and Liu, P. L. (2005). A numerical investigation of Boumerdes-Zemmouri (Algeria) earthquake and tsunami. Comput. Model. Eng. Sci., 10(2), 171.

Wang, X., and Liu, P. L. F. (2006). An analysis of 2004 Sumatra earthquake fault plane mechanisms and Indian Ocean tsunami. J. Hydraul. Res., 44(2), 147-154. doi: 10.1080/00221686.2006.9521671

Wang, X., and Liu, P. L. F. (2007). Numerical simulations of the 2004 Indian Ocean tsunamis-coastal effects. Journal of Earthquake and Tsunami, 1(03), 273-297. Wu, T. R., Chen, P. F., Tsai, W. T., and Chen, G. Y. (2008). Numerical study on tsunamis excited by 2006 Pingtung earthquake doublet. Terr. Atmos. Ocean. Sci. doi: 10.3319/TAO.2008.19.6.705(PT)

The following questions should be addressed: Q: Which formulas and parameters are used, in particular for bottom friction (Manning coefficient)? The bottom friction has an impact on the simulated tsunami amplitude at the coast.

A: We have added the description of Manning coefficient. [Page 6, lines 9-10]

Nonlinear shallow water equation for Cartesian coordinate is used:

[Figure]

$\partial\eta/\partial t + \partial P/\partial x + \partial Q/\partial y = -\partial h/\partial t$

$\partial P/\partial t + \partial/\partial x \{P^2/H\} + \partial/\partial y \{PQ/H\} + gH^*\partial\eta/\partial x + F\_x = 0$

$\partial Q/\partial t + \partial/\partial x \{PQ/H\} + \partial/\partial y \{Q^2/H\} + gH^*\partial\eta/\partial y + F\_y = 0$

$\eta$ is the free-surface displacement. P and Q are the horizontal volume discharges. g is gravity. h is the still water depth. H is the total water depth, H=$\eta$+h. Fx and Fy are the bottom frictions.

$F\_x = gn^2/H^{(7/3)}{}^*P(P^2+Q^2)^{(1/2)}$

$F\_y = gn^2/H^{(7/3)}{}^*Q(P^2+Q^2)^{(1/2)}$

n is Manning's roughness coefficient. In this study, Manning coefficient is 0.013, which represents a smooth surface (Wu, et al., 2008; Wang 2009).

Reference: Wang, X. (2009). User manual for COMCOT version 1.7 (first draft). Cornel University, 65. Wu, T. R., Chen, P. F., Tsai, W. T., and Chen, G. Y.: Numerical Study on Tsunamis Excited by 2006 Pingtung Earthquake Doublet, Terr. Atmos. Ocean. Sci., 19, 705-715, 2008. doi: 10.3319/TAO.2008.19.6.705(PT)

Q: Which bathymetry and topography data is used? Free GEBCO and SRTM?

A: We have added it. [Page 6, lines 10-11]

Q: The resolution of 1 minute for the inner mesh is quite rough for simulations that should give estimates of the tsunami amplitude at the coast. Our experience from hind casts of real events suggests that at the coast line, the horizontal resolution should be 500m (edge length in an unstructured triangular grid) or better. This should be transferable, as COMCOT also is a model with first order spatial discretization.

A:The Figure 1 presents the time series by uniform slip distribution at station 25 in different resolution of topography. The time series are similar. For resolution, 1 minute is better than 2 minute and for time spent, 1 minute is less than 30 arc-sec. Therefore,

to consider the resolution of simulation and time spent, the resolution of 1 minute was applied.

Q: Where are the tide gauges located? See also point 14, references. On the one hand, the exact location is not really important, because the study could be performed with virtual sensor locations or coastal forecast points, but – to reproduce the results, the locations of the (real or virtual) gauges are needed, – for hind casts of real events, the location and measurements from real tide gauges are needed, – the simulation of the tsunami wave form at a tide gauge that is located e.g., inside a harbor or narrow bight is very sensitive to errors in the representation of bathymetry and topography (1min resolution for sure is too coarse!) and to the choice of the roughness parameter (wave reflections). – The comparison in fig. 6 may be spoiled by different gauge locations. Distance to the source is not the only parameter, as it is also stated in the paper, too (e.g., page 7 line 23-24).

A: We have added location information and removed fitting line [Page 6, lines 13-18; Pages 20-21] We list the location of the gauges in the Table 1. The fittings of Fig. 6 just give a rough relationship between wave height and distance for the tsunami source which is perpendicular the coast line. Of course, the distance is not the only parameter for wave height attenuation. We agree to remove the fitting lines.

11. Are mathematical formulae, symbols, abbreviations and units correctly defined and used? If the formulae, symbols or abbreviations are numerous, are there tables or appendixes listing them? Q: Equation (1): W for width, L for length: It's obvious, but nevertheless should be added in the text above. Which value for $\mu$ is assumed when estimating Mw? And as a non-seismologist, I would like to ask if the estimate of D = 8.25m is really obvious? Section 2.2: Not my field of expertise at all.

A: We have done it. [Page 3, lines 21-23]

We will add the definition of symbols (W and L) in the text. $\mu$ usually sets 30GPa and it assumes that crust is elastically uniform. The estimation of slip and Mw is from

fault geometry and parameter assuming as $\mu$. We analyzed the relation between Mw and average slip (D) in Fig 2. The public finite fault slip models of global slip earthquakes are from the website (http://equake-rc.info/SRCMOD/). This figure appears the trend between Mw and average slip and its boundary. For Mw8.15, the range could be 200~1000 cm. It explains that our estimation, which follows the trend and in the possible boundary, is reasonable.

―――――――――――

[Figure]

[Figure]

**Fig. 1.** Fig. 1 The time series by uniform slip distribution at station 25 in different resolution of topography. Blue line is 2 minute, red line is 1 minute and yellow line is 30 arc-sec.

**Fig. 2.** Fig R1. Mw of real events and their average slips with 2 standard deviation (http://equake-rc.info/SRCMOD/). Open circles represent the inverse slip results in each study. Solid circles represent the

**Supplement:**

[revised manuscript text omitted]

---

## Author Comment (AC2) · 12 Feb 2018

We would like to thank Reviewer for the detailed revision and important suggestions. We improved the manuscript following suggestions. The attachment is modified manuscript.

Q: Page 2, lines 3-5. If the earthquakes associated to the historic tsunamis mentioned in the text have any magnitude estimation, please provide the value and include the reference. For instance, the 1867 tsunami, magnitude?

A: We have done it. [Page 2, lines 3-4] The 1867 Keelung earthquake was inferred approximately Mw 7.0 (Tsai 1985; Ma and Lee 1997; Cheng et al., 2016; Yu et al., 2016).

Reference:

Cheng, S. N., Shaw, C. F., and Yeh, Y. T. (2016). Reconstructing the 1867 Keelung Earthquake and Tsunami Based on Historical Documents. Terr. Atmos. Ocean. Sci., 27(3). doi:10.3319/TAO.2016.03.18.01(TEM)

Ma, K. F., and Lee, M. F. (1997). Simulation of historical tsunamis in the Taiwan region. Terr. Atmos. Ocean. Sci., 8(1), 13-30. doi: 10.3319/TAO.1997.8.1.13(T) Tsai, Y. B. (1985). A study of disastrous earthquakes in Taiwan, 1683–1895. Bull. Inst. Earth Sci. Acad. Sin, 5, 1-44.

Yu, N.-T., Yen, J.-Y., Chen, W.-S., Yen, I. C., and Liu, J.-H.: Geological records of western Pacific tsunamis in northern Taiwan: AD 1867 and earlier event deposits, Mar. Geol., 372, 1-16, 2016. doi:10.1016/j.margeo.2015.11.010

Q: Page 2, lines 13-15. When comparing PTHA and PSHA, authors mentioned in the text that PSHA works with ground-motion parameters. So, can you complete the idea by specifying that PTHA works with tsunami wave amplitudes, or some other wave measurements? If there is any reference, please include it.

A: We have done it. [Page 2, lines 14-19]

Geist and Parsons (2006) mentions that the tsunami wave amplitudes follow a definable frequency-size distribution over a sufficiently long amount of time at a given coastal region (Soloviev, 1969; Houston et al., 1977; Horikawa and Shuto, 1983; Burroughs and Tebbens, 2005). This method is of great use in establishing tsunami probability for regions if there is an extensive catalog of observed tsunami wave heights (Geist and Parsons, 2006). The other approach is numerical simulation (Geist, 2002; Geist and Parsons, 2006; Geist and Parsons, 2009) which applies the stochastic slip model to

estimate the tsunami amplitudes probability as this study.

Reference:

Burroughs, S.M., Tebbens, S.F. (2005). Power law scaling and probabilistic forecasting of tsunami runup heights. Pure Appl. Geophys. 162, 331–342

Geist, E.L., (2002). Complex earthquake rupture and local tsunamis. J. Geophys. Res. 107. doi:10.1029/2000JB000139.

Geist, E. L., and Parsons, T. (2006). Probabilistic analysis of tsunami hazards. Natural Hazards, 37(3), 277-314. doi 10.1007/s11069-005-4646-z

Geist, E. L., and Parsons, T. (2009). Assessment of source probabilities for potential tsunamis affecting the US Atlantic coast. Marine Geology, 264(1), 98-108.

Horikawa, K. and Shuto, N. (1983). Tsunami disasters and protection measures in Japan, In: K. Iida and T. Iwasaki (eds), Tsunamis-Their Science and Engineering, Terra Scientific Publishing Company, pp. 9–22.

Houston, J. R., Carver, R. D. and Markle, D. G. (1977). Tsunami-wave elevation frequency of occurrence for the Hawaiian Islands. Technical Report H-77-16, U.S. Army Engineer Waterways Experiment Station, Vicksburg, MS, 66 pp.

Soloviev, S. L. (1969). Recurrence of tsunamis in the Pacific. In: W. M. Adams (ed.), Tsunamis in the Pacific Ocean, East-West Center Press, pp. 149–163.

Q: Page 3. Line 2. Please, provide the reference for the magnitude range, Mw 7.5-8.7.

A: We have done it. [Page 3, line 5]

Reference:

Hsu, Y. J., Ando, M., Yu, S. B., and Simons, M. (2012). The potential for a great earthquake along the southernmost Ryukyu subduction zone. Geophysical Research Letters, 39(14).

СЗ

Q: Page 3. Line 12. About the fault geometry setting. Which is the source depth of the top (or bottom) of the fault plane? I think it has not been specified yet in the text.

A: We have done it. [Page 3, lines 14-15]

The fault geometry setting refers to Hsu et al. (2012) and fault model extends from the Ryukyu Trench to a depth of 13 km.

Q: Page 3. Line 15, please complete to "...in dip slip faults".

A: Thank you. We have done it. [Page 3, line 18]

Q: Page 3. Eq. (1), please, specify what is L, and W.

A: We have done it. [Page 3, line 21]

Q: Page 3. Line 18. I suggest to change "constant" by "parameter". Strictly speaking, in elastic heterogeneous media, the Lamè parameters (lambda and mu) vary in space.

A: We have done it. [Page 3, line 22]

Q: Page 3. In Eq. (2). Which is the value assumed for mu?

A: We have done it. [Page 3, lines 22-23]

Q: Page 3. Section 2.1. When the authors compute the earthquake magnitude, average slip and fault area. Did the authors compare (or contrast) these values with any magnitude/fault-size scaling relationship for subduction earthquakes? It could be interesting to compare these values with any magnitude/size scaling relationship for subduction zones.

A: We analyzed the relation between Mw and average slip (D) in Figure 1. The public finite fault slip models of global slip earthquakes are from the website (http://equake-rc.info/SRCMOD/). This figure appears the trend between Mw and average slip and its boundary. For Mw8.15, the range could be  $200 \sim 1000$  cm. It explains that our estimation, which follows the trend and in the possible boundary, is reasonable.

Reference:

ChiChi (1999): Ma et al. (2000); Chi et al. (2001); Zeng and Chen (2001); Wu et al. (2001); Zhang et al. (2004)

Tohoku (2011): Ammon et al. (2011); Ide et al. (2011); Lay et al. (2011); Shao et al. (2011); Yagi and Fukahata (2011); Yamazaki et al. (2011); Wei et al. (2012)

Maule (2010): Delouis et al. (2010); Hayes (2010); Shao et al. (2010); Sladen (2010); Luttrell et al. (2011)

Sumatra (2004): Ammon et al. (2005); Ji (2005); Rhie et al. (2007)

Sumatra (2012): Hayes (2012); Shao et al. (2012); Wei (2012); Yue et al. (2012)

Tokachi-Oki (2003): Yamanaka and Kikuchi (2003); Koketsu et al. (2004); Tanioka et al. (2004); Yagi (2004)

Tocopilla (2007): Ji (2007); Sladen (2007); Zeng et al. (2007); Béjar-Pizarro et al. (2010); Motagh et al. (2010)

Reference:

Ammon, C. J., J. Chen, H.-K. Thio, D. Robinson, S. Ni, V. Hjorleifsdottir, H. Kanamori, T. Lay, S. Das, D. Helmberger, G. Ichinose, J. Polet, and D. Wald. (2005). Rupture process of the great 2004 Sumatra-Andaman earthquake, Science, 308, 1133-1139.

Ammon, C. J., T. Lay, H. Kanamori, and M. Cleveland (2011) A rupture model of the 2011 off the Pacific coast of Tohoku Earthquake, Earth Planets Space, 63, 693–696.

Bejar-Pizzaro M., Carrizo D., Socquet A., Armijo R., (2010) Asperities, barriers and transition zone in the North Chile seismic gap: State of the art after the 2007 Mw 7.7 Tocopilla earthquake inferred by GPS and InSAR data, Geoph. Journ. Int., GJI-S-09-0648, doi: 10.1111/j.1365-246X.2010.04748.x

Chi, W. C., D. Dreger, and A. Kaverina. 2001. Finite-source modeling of the 1999

Taiwan (Chi-Chi) earthquake derived from a dense strong-motion network. Bull. Seis. Soc. Am 91 (5):1144-1157.

Delouis B., J. M. Nocquet, M. Vallée (2010). Slip distribution of the February 27, 2010 Mw = 8.8 Maule Earthquake, central Chile, from static and high-rate GPS, InSAR, and broadband teleseismic data, Geophys. Res. Lett., 37, L17305, doi:10.1029/2010GL043899.

G., (NEIC. Maule 2010) Updated Hayes Result of Mw 8.8 Chile the Feb 27, 2010 Maule, Earthquake, http://earthquake.usgs.gov/earthquakes/eqinthenews/2010/us2010tfan/finite\_fault.php, last accessed August 19, 2013.

Hayes G., (NEIC, Sumatra 2012) Preliminary Result of the Apr 11, 2012 Mw 8.6 Earthquake Off the West Coast of Northern Sumatra, http://earthquake.usgs.gov/earthquakes/eqinthenews/2012/usc000905e/finite\_fault.php, last accessed August 19, 2013.

Ide S., A. Baltay, and G. C. Beroza (2011). Shallow Dynamic Overshoot and Energetic Deep Rupture in the 2011 Mw 9.0 Tohoku-Oki Earthquake, 332, 1426-1429, DOI: 10.1126/science.1207020

Ji, C. (2005). Preliminary Rupture Model for the December 26, 2004 earthquake, off the west coast of northern Sumatra, magnitude 9.1, http://neic.usgs.gov/neis/eq\_depot/2004/eq\_041226/neic\_slav\_ff.html

Ji C. (UCSB, Tocopilla 2007) Preliminary Result of the Nov 14, 2007 7.81 ANTOFAGASTA, CHILE Mw Earthquake, http://www.geol.ucsb.edu/faculty/ji/big earthquakes/2007/11/anto/anto.html, last accessed August 11, 2013.

Koketsu, K., K. Hikima, S. Miyazaki, and S. Ide. (2004). Joint inversion of strong motion and geodetic data for the source process of the 2003 Tokachi-oki, Hokkaido,

earthquake. Earth Planets and Space 56 (3):329-334.

Lay T., C. J. Ammon, H. Kanamori, L. Xue, and M. J. Kim (2011). Possible large neartrench slip during the 2011 Mw 9.0 off the Pacific coast of Tohoku Earthquake. Earth Planets Space. 63, 687–692.

Luttrell, K. M., Tong, X., Sandwell, D. T., Brooks, B. A., and Bevis, M. G. (2011). Estimates of stress drop and crustal tectonic stress from the 27 February 2010 Maule, Chile, earthquake: Implications for fault strength. Journal of Geophysical Research: Solid Earth (1978–2012), 116(B11).

Ma, K. F., T. R. A. Song, S. J. Lee, and H. I. Wu. (2000). Spatial slip distribution of the September 20, 1999, Chi-Chi, Taiwan, earthquake (M(W)7.6) - Inverted from teleseismic data. Geophys. Res. Lett. 27 (20):3417-3420.

Motag, M., B. Schurr, J. Anderssohn, B. Cailleau, T. R. Walter, R. Wang, J.-P. Villotte, (2010) Subduction earthquake deformation associated with 14 November 2007, Mw 7.8 Tocopilla earthquake in Chile: Results from InSAR and aftershocks, Tectonophysics 490, 60–68

Rhie, J., D. Dreger, R. Burgmann, and B. Romanowicz. (2007). Slip of the 2004 Sumatra–Andaman Earthquake from joint inversion of long-period global seismic waveforms and GPS static Offsets, Bull. Seismo. Soc. Am., 97(1A): S115–S127.

Shao, G., X. Li and C. Ji. (UCSB, sumatra 2012). Preliminary Result of the Apr 11, 2012 Mw 8.64 sumatra Earthquake, http://www.geol.ucsb.edu/faculty/ji/big\_earthquakes/2012/04/10/sumatra.html, last accessed August 19, 2013.

Shao, G., X. Li, C. Ji. and T. Maeda (2011). Focal mechanism and slip history of 2011 Mw 9.1 off the Pacific coast of Tohoku earthquake, constrained with teleseismic body and surface waves, Earth Planets Space, 63 (7), 559-564.

Shao, G., X. Li, Q. Liu, X. Zhao, T. Yano and C. Ji(UCSB, Maule 2010).Pre-

**C7**

liminary slip model of the Feb 27, 2010 Mw 8.9 Maule, Chile Earthquake, http://www.geol.ucsb.edu/faculty/ji/big\_earthquakes/2010/02/27/chile\_2\_27.html, last accessed September 24,2013.

Sladen A. (Caltech, Tocopilla 2007). Preliminary Result 11/14/2007 (Mw 7.7), Tocopilla Earthquake, Chile. Source Models of Large Earthquakes. http://www.tectonics.caltech.edu/slip\_history/2007\_tocopilla/tocopilla.html, last accessed July 1, 2013.

Maule 2010). Sladen Α. (Caltech, Preliminary Result, 02/27/2010 Chile. Source of Large (Mw 8.8), Models Earthquakes. http://www.tectonics.caltech.edu/slip\_history/2010\_chile/index.html

Tanioka, Y., K. Hirata, R. Hino, and T. Kanazawa. (2004). Slip distribution of the 2003 Tokachi-oki earthquake estimated from tsunami waveform inversion. Earth Planets and Space 56 (3):373-376.

Wei Sumatra April/11/2012 S. (Caltech, 2012). (Mw Earthquakes. 8.6), Sumatra. Source Models of Large http://www.tectonics.caltech.edu/slip\_history/2012\_Sumatra/index.html, last accessed July 1, 2013.

Wei, S. J., R.W. Graves, D. Helmberger, J.P. Avouac and J.L. Jiang (2012) Sources of shaking and flooding during the Tohoku-Oki earthquake: A mixture of rupture styles, Earth and Planetary Science Letters, 333-334, 91-100.

Wu, C. J., M. Takeo, and S. Ide. (2001). Source process of the Chi-Chi earthquake: A joint inversion of strong motion data and global positioning system data with a multifault model. Bull. Seis. Soc. Am 91 (5):1128-1143.

Yagi, Y. (2004). Source rupture process of the 2003 Tokachi-oki earthquake determined by joint inversion of teleseismic body wave and strong ground motion data, Earth Planets Space, 56, 311–316.

Yagi, Y. and Fukahata, Y., (2011). Rupture process of the 2011 Tohoku-oki earthquake and absolute elastic strain release, Geophys. Res. Lett, 38, L19307, doi:10.1029/2011GL048701.

Yamanaka, Y., and M. Kikuchi. (2003). Source process of the recurrent Tokachi-oki earthquake on September 26, 2003, inferred from teleseismic body waves. Earth Planets and Space 55 (12):E21-E24.

Yamazaki, Y., T. Lay, K. F. Cheung, H. Yue, and H. Kanamori (2011). Modeling near-field tsunami observations to improve finite-fault slip models for the 11 March 2011 Tohoku earthquake, Geophys. Res. Lett., 38, L00G15, doi:10.1029/2011GL049130.

Yue, H, T. Lay and K. D. Koper (2012), En Echelon andOrthogonal Fault Ruptures of the 11 April 2012 Great Intraplate Earthquakes. Nature, 490, 245-249, doi:10.1038/nature11492.

Zeng, Y. H., and C. H. Chen. (2001). Fault rupture process of the 20 September 1999 Chi-Chi, Taiwan, earthquake. Bull. Seis. Soc. Am 91 (5):1088-1098.

Zeng, Y., G.Hayes and C. Ji (2007; USGS, Online Model). Preliminary Result of the Nov 14, 2007 Mw 7.7 Antofagasto, Chile Earthquake, http://earthquake.usgs.gov/earthquakes/eqinthenews/2007/us2007jsat/finite\_fault.php, last accessed August 20, 2013.

Zhang, W., T. Iwata, K. Irikura, A. Pitarka, and H. Sekiguchi (2004), Dynamic rupture process of the 1999 Chi-Chi, Taiwan, earthquake, Geophys. Res. Lett., 31, L10605, doi:10.1029/2004GL019827.

Q: Page 3, line 25. For completeness purposes, please provide the scalar seismic moment, M0 for the corresponding Mw 8.15.

A: We have done it. [Page 4, line 2]

Q: Page 4. Please clarify or complete the sentence in line 8, because there is a dot at the end of the sentence, so it is not clear what Eq. (4) means or represents. The 2D Fourier spectrum amplitude of what?

A: We have done it. [Page 4, lines 9-11]

Eq. (4) illustrate that the spectrum of static slip distribution in wavenumber domain is following k-2 decay. In Eq. (4), Dx,y is slip distribution and its spectrum is proportional to k-2. Andrews (1980) derived the k-2 from the relationship of slip and stress change.

Q: Page 4. Line 10. Please, to be consistent with the notation in Eq. (4), please clarify the meaning of "F", or, change F by Fs,t which represents the 2D discrete Fourier transform of Dx,y. Also, for completeness purposes, specify that Dx,y is the slip distribution over a 2D lattice, for instance.

A: Thank you. We have done it. [Page 4, line 13]

Q: Page 4. In line 10, please complete, "...wave number.", by "...radial wavenumber."

A: Thank you. We have done it. [Page 4, line 13]

Q: Page 4. Line 13, please correct "corner frequency" by "corner radial wavenumber", because kc is not a frequency.

A: Thank you. We have done it. [Page 4, line 16; Page 5, line 7 and Page 15, line 13]

Q: Page 4. Line 14. What happen with the phase beyond kc? Please, clarify. Or, the last sentence "Within the kc,....(Geist, 2002)." could be deleted because authors are describing the overall characteristics of the slip and not describing the details of how the random slip is generated numerically in the practice.

A: We have removed this sentence. Beyond the corner radial wavenumber, kc, the slip spectrum decays with k-2. The generation of random slip is explained in next paragraph, Page 5.

Q: Page 4. Eq. (5). Please, be careful and clear with the mathematical notation. What does FËĘ(-1) represent ?. Is it the inverse 2D discrete Fourier transform?

A: Thank you. We have done it. [Page 4, lines 22-23]

Q: Page 4. Line 23. Please, specify that PDF is Probability Density Function, I think it has not been mentioned before in the text.

A: Thank you. We have done it. [Page 4, line 27]

PDF is Probability Density Function.

Q: Page 5. Line 3. Complete the units in the sentence, "...5x5 km...", by "...5x5 kmËĘ2...".

A: Thank you. We have done it. [Page 5, line 8; Page 5, line 32; Page 16, line 4]

Q: Page 5. Line 3. Please, clarify that 24x14 are along strike and dip respectively.

A: Thank you. We have done it. [Page 5, line 8]

Page 5. Line 1-4. I will ask the authors to provide some details about how the stochastic slip distribution is generated, and to be clear on the choice of parameters and discuss about the results. Please, read the following comments.

Q: The authors used the values of the Levy PDF suggested by Lavallee et al. (2006), so please clarify in the manuscript that those values were estimated from a stochastic 2D model in the dip slip direction, obtained for the Northridge earthquake. So, why do you use parameters from a shallow crustal earthquake occurred in California to characterize a interplate subduction zone earthquake? Please justify, or discuss.

A: Thank you. We have done it. [Page 5, lines 10-15]

Furthermore, in this study, we do not focus on the values of characteristic for different kinds of faults. Therefore, we decided to simply apply these values which had been published already.

**Reference:**

Davis, T. L. (1994). 1994 Northridge earthquake. Nature, 372, 167.

Q: Notice that according to Lavallee et al (2006) and others, the scaling exponent is (nu+1) so, the Power Spectrum Density of slip is, P(k) âLij kËE(-(nu+1)), it implies that the slip spectrum behaves as, D(k) âLij kËE(-(nu+1)/2). The authors generate random variables using the Levy distribution, and imposed P(k) âLij kËE(-2) as shown in Fig. 1c, so, the slip in the wavenumber domain behaves as, D(k) aLij kEE(-1), and Figure 1 is ok, but the slip spectrum does not follow the kEE(-2) source characteristic discussed at the beginning of Section 2.2. Please, clarify this point in the text. Also, discuss the effect in the spatial distribution of slip of this choice (falloff as kEE(-1) of the slip spectrum amplitude in the wavenumber domain), versus a slip spectrum that falloff as kEE(-2). From the results shown in Fig. 1, authors generated a slip spectrum that decays as  $k \ddot{E} \xi$ (-1) because they imposed the power spectrum density as P(k) âĹij kËĘ(-2), but in the legend they say "This slip spectrum decays with exponent of -2 and...", so, it is an inconsistency for me. Please, be clear on the choice, and the terminology used when generating spatial random fields. Herrero & Bernard (1994), Andrews (1981), and others, used a stochastic slip model with a 2D Fourier spectrum that decays as kEE-2 which means, D(k) âLij kEE(-2). I am not saying the authors are wrong in their choice, it is only that some parts of the text need some clarification, justification of the choice, or discussion about the assumptions done.

A: We are very sorry for the confusion. In general, the spectrum of slip distribution is proportional to k-2 (Herrero and Bernard 1994; Andrews 1980; Tsai 1997). ( $|D(k)| \sim k^{(-nu-1)}$ , nu=1) At the beginning of Section 2.2, the Eq. (1) wants to present the spectrum of slip distribution is proportional to k-2. Fig. 1c shows slip spectrum and it consist with k-square. In Lavallee et al (2006), it is formularized by power spectrum density so that there is a disparity of square. We have modified the sentence and Fig. 1c. [Page 4, lines 9-11; Page 15]

Q: Page 5. Line 3. Why did you set a 5x5 subfault size? Did you test different subfault sizes?

A: For 5x5 km2, the resolution of 1 minute ( $\sim$ 1.8 km) should be enough to calculate and differentiate the surface deformation.

Q: Page 5. Line 3. Did you assume a constant slip at each subfault? If it is the case, how do you treat the non-smooth slip boundary condition at the boundaries of the fault? Did you apply a taper at all the borders, if not, authors should discuss or justify their treatment?

Q: Page 5. Lines 15-19. Same comment as done in Page 5, line 3, about the assumption of uniform slip at each subfault.

A: Thank you. We have done it. [Page 5, lines 22-27]

In this study, we do not do any smooth for slip distribution or its boundary. They are complete uniform slip and stochastic process over the fault model. There are two reasons for this application. The first is that we do not have information for where is locked or the location of asperity often repeats in historical event. The second is that there are some studies present the asperity expanding to the boundary of fault model (lde et al., 2011; Lay et al., 2011; Shao et al., 2011; Yue and Lay 2011). According to these, we do not prefer to apply any extra constraint. If we have more information about the characteristic of rupture behavior for this region, we would consider giving a constraint.

Reference:

Ide, S., Baltay, A., and Beroza, G. C. (2011). Shallow dynamic overshoot and energetic deep rupture in the 2011 Mw 9.0 Tohoku-Oki earthquake. Science, 332(6036), 1426-1429. doi: 10.1126/science.1207020

Lay, T., Ammon, C. J., Kanamori, H., Xue, L., amd Kim, M. J. (2011). Possible large near-trench slip during the 2011 Mw 9.0 off the Pacific coast of Tohoku Earthquake.

Earth, planets and space, 63(7), 32. doi:10.5047/eps.2011.05.033

Shao, G., Li, X., Ji, C., and Maeda, T. (2011). Focal mechanism and slip history of the 2011 Mw 9.1 off the Pacific coast of Tohoku Earthquake, constrained with teleseismic body and surface waves. Earth, planets and space, 63(7), 9. doi:10.5047/eps.2011.06.028

Yue, H., and Lay, T. (2011). Inversion of high-rate (1 sps) GPS data for rupture process of the 11 March 2011 Tohoku earthquake (Mw 9.1). Geophysical Research Letters, 38(7). doi: 10.1029/2011GL048700

Q: Page 5, line 15. I would suggest to use "computational domain" instead of "...numerical model".

A: Thank you. We have done it. [Page 5, line 32]

Q: Page 5. Line 15. Complete the units in 5x5 kmËĘ2.

A: Thank you. We have done it. [Page 5, line 32].

Q: Page 5, lines 21-25. Why do you use 4 min and 1 min for the nested grids? Did you test a different grid size? Which bathymetry/topography is used in the numerical simulation of the tsunami? Please include a reference. For instance, GEBCO (https://www.gebco.net/) provides a global 30 arc-sec bathymetry, which has a better resolution than the bathymetry used in this work. Please comment on it. Which is the boundary condition set at the coastlines (the boundary between wet and dry domains)?. Do you assume a vertical wall condition, or do you allow inundation? Did you impose any friction, if yes, which one is the Manning's coefficient used in the simulation?

A: Thank you. We have done it. [Page 6, lines 11-12; Page 6, lines 13-18]

NOAA's open data is used. It is free GEBCO and SRTM. The data can be download from: https://maps.ngdc.noaa.gov/viewers/wcs-client/ The Figure 2 presents the time

series by uniform slip distribution at station 25 in different resolution of topography. The time series are similar. For resolution, 1 minute is better than 2 minute and for time spent, 1 minute is less than 30 arc-sec. Therefore, to consider the resolution of simulation and time spent, the resolution of 1 minute was applied. COMCOT is capable of efficiently studying the entire life-span of a tsunami, including its generation, propagation, runup and inundation. COMCOT also supports the nested grid system that the finer grid can be placed on a coarser grid to increase the resolution locally (Wang 2009). In this study, Manning coefficient is 0.013, which represents a smooth surface (Wu, et al., 2008).

**Reference:**

Wang, X. (2009). User manual for COMCOT version 1.7 (first draft). Cornel University, 65.

Wu, T. R., Chen, P. F., Tsai, W. T., and Chen, G. Y.: Numerical Study on Tsunamis Excited by 2006 Pingtung Earthquake Doublet, Terr. Atmos. Ocean. Sci., 19, 705-715, 2008. doi: 10.3319/TAO.2008.19.6.705(PT)

Q: Page 6. Sentence in line 5-6 is a bit confusing, please rephrase to clarify.

A: We are very sorry for confusing. We have done it. [Page 6, lines 29-31]

Q: Page 6. Section 3.1. If I understand, authors used the vertical seafloor displacement as initial condition to propagate the tsunami, and the horizontal motion of the seabed is not included in the simulation. I will suggest to clarify better these assumptions in Section 3.1.

A: We have done it. [Page 6, lines 29-31]

Q: Page 7. Section 3.3. Authors say basically that they computed the probability of the PTA by histograms, but from my understanding they show (Fig. 5) a probability density estimated from the numerical PTA data. I think authors could say/argue a little bit more about this, in terms of this choice and analysis. I mean, does the data follow

any distribution (e.g. Gaussian, Levy, Log-normal)? Are the PTA data (simulated) Gaussian distributed? Is it possible to estimate the probability of exceeding a certain input value from these numerical results? I think some of these aspect is not discussed or mentioned in the text.

A: We have added in the text. [Page 8, lines 11-13]

Q: Page 7. Line 11. Please complete the idea that after generating the second set of slip models, the tsunami is simulated.

A: We have done it. [Page 8, lines 2-6] The histograms, first set, and black lines, second set, are similar. The second set illustrate that the PTA distribution by 100 times tsunami simulations is approximately reliable.

Q: Page 7. Paragraph 3. When you compare PTA versus distance, how do you define or measure the distance between source and station? At least, it could be mentioned or discussed in the text.

A: We have done it. [Page 8, line 22]

Q: Page 8. Lines 14-16. Please, provide the references for the Maule, Tohoku and Sumatra earthquakes. A: We have done it. [Page 9, line 6-8]

Reference:

Chile earthquake (Lay et al., 2010; Fritz et al., 2011)

Lay, T., Ammon, C. J., Kanamori, H., Koper, K. D., Sufri, O., & Hutko, A. R. (2010). Teleseismic inversion for rupture process of the 27 February 2010 Chile (Mw 8.8) earthquake. Geophysical Research Letters, 37(13).

Fritz, H. M., Petroff, C. M., Catalan, P. A., Cienfuegos, R., Winckler, P., Kalligeris, N., Weiss, R., Barrientos, S. E., Meneses, G., Valderas-Bermejo, C., Ebeling, C., Papadopoulos, A., Contreras, M., Almar, R., Dominguez, J. C., and Synolakis, C. E. (2011). Field survey of the 27 February 2010 Chile tsunami. Pure and Applied Geophysics, 168(11), 1989-2010.

Tohoku earthquake (Goda et al., 2015; Goda and Song, 2016)

Goda, K., and Song, J. (2016). Uncertainty modeling and visualization for tsunami hazard and risk mapping: a case study for the 2011 Tohoku earthquake. Stochastic Environmental Research and Risk Assessment, 30(8), 2271-2285.

Goda, K., Yasuda, T., Mori, N., and Mai, P. M. (2015). Variability of tsunami inundation footprints considering stochastic scenarios based on a single rupture model: application to the 2011 Tohoku earthquake. Journal of Geophysical Research: Oceans, 120(6), 4552-4575.

Sumatra earthquake (Lay et al., 2005)

Lay, T., Kanamori, H., Ammon, C. J., Nettles, M., Ward, S. N., Aster, R. C., ... & DeShon, H. R. (2005). The great Sumatra-Andaman earthquake of 26 December 2004. Science, 308(5725), 1127-1133.

Q: Page 8. Lines 22-31. The results discussed here are obtained at several sites, but It is not clear where the sites (tides gauges) are exactly located, right at the boundary, or surrounded by a wet domain even during the tsunami evolution? If the latter is true, the comparison of maximum tsunami wave height (this study) is not exactly straightforward comparable to runup (analyzed in other studies). Also, authors should comment on the effect (or limitations) of the grid resolution (1 arc-min, used in this study) over the results obtained. I suspect this coarse grid may have an effect on the simulations near the coast.

A: Thank you. These stations are surrounded by a wet domain so that we have modified this part. [Page 9, lines 13-22]

In comment of Page 5, lines 21-25, we provide a test in different resolution of topography to prove that the resolution of 1 minute can be accepted.

Q: Page 8, line 17. Clarify what "lecture" means.

A: We have done it. [Page 9, line 9]

Q: Page 9. Line 29. I would suggest to complete the idea in the sentence, "Furthermore, interpolation has a tremendous effect for the exponent value becoming larger with grid size reducing (Tsai, 1997).", because it refers to how the exponent and correlation lengths are computed from the solutions of slip models of earthquakes. On the other hand, some authors assume k-2 slip models based on other physical considerations.

A: We have done it. [Page 10, lines 18-20]

Interpolation for a given geometry will affect the exponent of k. For example, the exponent value of the original slip model of the Northridge earthquake from Zeng and Anderson (1996) is 1.876 in Tsai (1997). The slip model is interpolated by making the dimension of the element size one-half of the original size (0.5x0.5 km2). The slip distribution is smoothed by the interpolation and the new exponent value is 3.767. The exponent value is 4.202 when the slip model is interpolated by making the dimension of the element size one-fourth of the original size. Our point from mathematical operation is that interpolation make original pattern smoother as a filter depresses the short wavenumber and enhancing the long wavenumber.

Q: Figure 1. Clarify units, X? k? length km or 5km? To avoid misunderstanding, I suggest to delete the label "Northrigde earthquake" in the Fig 1a, and you can mention it in the caption (e.g. Levy parameters were taken from Lavallee et al.....obtained for the Northridge earthquake.), because the realization shown is for an Mw 8.15 earthquake and not for the Northridge earthquake. Fault axis along dip and strike are confusing too. I will suggest to plot the real distance along strike and dip directions (with the correct units) and not the "indexes" of each subfault. What do represent the colorbar?

See my comments about P(k) and D(k), what is shown in Fig 1c is not what is written in the caption.

A: We have done it. [Page 15]

X is random variable (the filtered slip) so that the unit of X is meter. The unit of k is km-1 ( $(kx^2+ky^2)^{-2}$ ).

Q: Figure 2. I suggest to contextualize at the beginning the region of the study area, (e.g. Map of Taiwan...for example). Correct 5x5 km by 5x5 kmËĘ2. Is the white box the nested inner grid? Colorbar?

A: We have done it. [Page 16] The colorbar presents the elevation in km.

Q: Figure 3. I would suggest specify that the "energy propagation" corresponds to, maximum tsunami wave height, for instance. Colorbar?

A: We have done it. [Page 17, lines 1-2] The colorbar presents the maximum tsunami wave height in meter (b, d, and f).

====== Tables ====== Q: Table 1. The description of the table and caption is a bit confusing. What is the meaning of Max(uni)? A suggestion is that a part of the description given at the end of the table can be moved to the caption, and authors can put the units [m] directly beneath each variable description.

A: We have modified it. [Pages 20-21] Max(uni) means the maximum wave height in uniform slip case.

Please also note the supplement to this comment: https://www.nat-hazards-earth-syst-sci-discuss.net/nhess-2017-336/nhess-2017-336-AC2-supplement.pdf

**Fig. 1.** Mw of real events and their average slips with 2 standard deviation. Open circles represent the inverse slip results in each study. Solid circles represent the mean slip of each study for same event.

**Fig. 2.** The time series by uniform slip distribution at station 25 in different resolution of topography. Blue line is 2 minute, red line is 1 minute and yellow line is 30 arc-sec.

---

## Author Comment (AC3) · 12 Feb 2018

Sorry. The captions have some problems so that we re-upload them.

[Figure]

[Figure]

**Fig. 1.** The time series by uniform slip distribution at station 25 in different resolution of topography. Blue line is 2 minute, red line is 1 minute and yellow line is 30 arc-sec.

[Figure]

**Fig. 2.** Mw of real events and their average slips with 2 standard deviation. Open circles represent the inverse slip results in each study. Solid circles represent the mean slip of each study for same event.

---

## Author Response (AR1)

Re: (nhess-2017-336) Assessment of peak tsunami amplitude associated with a great earthquake occurring along the southernmost Ryukyu subduction zone for Taiwan region *by* Yu-Sheng Sun, Po-Fei Chen, Chien-Chih Chen, Ya-Ting Lee, Kuo-Fong Ma, and Tso-Ren Wu

Dear Prof. Lionello,

Thank you for reviewing this paper. We have made the revision to our manuscript intensively and reply the comments from reviewers carefully for your further consideration on the publication in Natural Hazards and Earth System Sciences (*NHESS*).

The authors highly appreciate the support of publication in *NHESS* from the reviewers and their helpful suggestion as well. We have made substantive modifications according to their suggestion. We deeply appreciate their suggestion, which has made the manuscript become much better. The annotated responses to the reviewers' comments and the details about our changes in the revised version of our manuscript are made accordingly in the files.

Attached please also find the electronic files of the revised manuscript for your further consideration of publication in *NHESS*. In the revised version, all modifications were marked in red for your reference. Any problem raised please let me know. Thank you very much.

With Best Regards, Yu-Sheng Sun

**Response (in black) to the comments of Reviewer (in blue)**

**Reviewer #1:**

**3. Are these up to international standards?**

4. Are the scientific methods and assumptions valid and outlined clearly?

7. Is the description of the data used, the methods used, the experiments and calculations made, and the results obtained sufficiently complete and accurate to allow their reproduction by fellow scientists (traceability of results)?

Concerning the application of the tsunami model: No. COMCOT is used as a black box. My major criticism is that the model is not validated for this area, and I strongly suggest to add a hind cast of a real event to prove that COMCOT with the chosen settings delivers realistic simulations. Probably, the last near field tsunami in 1867 is not well suited for a hind cast due to the lack of measurements, but the Tohoku tsunami 2011 should be a good test case also for Taiwan.

We have added some references in text. [Page 6, lines 2-6]

To solve the time dependent tsunami propagation, we adopt a well-validated numerical model, COMCOT (Cornell Multi-grid Coupled Tsunami Model). COMCOT is able to solve both linear and nonlinear shallow water equations on a Cartesian or Spherical coordinate systems (Wang 2009). In terms of validation, COMCOT has been widely used in studying many historical tsunami events, such as 1960 Chilean tsunami (Liu et al., 1995), 1992 Flores Islands tsunami (Liu et al., 1995), 2003 Algeria tsunami (Wang and Liu, 2005), 2004 Indian Ocean tsunami (Wang and Liu, 2006, 2007), and 2006 Ping-Tung tsunami (Wu, et al., 2008; Chen, et al., 2008). Taking the explicit leap-frog scheme to solve shallow water equation, COMCOT has the 2nd order accuracy in both special and time domains. COMCOT also supports the nested grid system that the finer grid can be placed on a coarser grid to increase the resolution locally. Thus, we can use finer grid in near-shore region and coarser grid in deep sea region.

**Reference:**

- Chen, P. F., Newman, A. V., Wu, T. R., and Lin, C. C. (2008). Earthquake Probabilities and Energy Characteristics of Seismicity Offshore Southwest Taiwan. Terr. Atmos. Ocean. Sci., 6, 697-703, doi: 10.3319/TAO.2008.19.6.697(PT)
- Liu, P. L. F., Cho, Y. S., Yoon, S. B., and Seo, S. N. (1995). Numerical simulations of the 1960 Chilean tsunami propagation and inundation at Hilo, Hawaii. In Tsunami: Progress in prediction, disaster prevention and warning (pp. 99-115). Springer,

Dordrecht. https://doi.org/10.1007/978-94-015-8565-1 7

- Liu, P. L. F., Cho, Y. S., Briggs, M. J., Kanoglu, U., and Synolakis, C. E. (1995). Runup of solitary waves on a circular island. J. Fluid Mech., 302, 259-285. doi: 10.1017/S0022112095004095
- Wang, X. (2009). User manual for COMCOT version 1.7 (first draft). Cornel University, 65.
- Wang, X., and Liu, P. L. (2005). A numerical investigation of Boumerdes-Zemmouri (Algeria) earthquake and tsunami. Comput. Model. Eng. Sci., 10(2), 171.
- Wang, X., & Liu, P. L. F. (2006). An analysis of 2004 Sumatra earthquake fault plane mechanisms and Indian Ocean tsunami. J. Hydraul. Res., 44(2), 147-154. doi: 10.1080/00221686.2006.9521671
- Wang, X., & Liu, P. L. F. (2007). Numerical simulations of the 2004 Indian Ocean tsunamis—coastal effects. Journal of Earthquake and Tsunami, 1(03), 273-297.
- Wu, T. R., Chen, P. F., Tsai, W. T., & Chen, G. Y. (2008). Numerical study on tsunamis excited by 2006 Pingtung earthquake doublet. Terr. Atmos. Ocean. Sci. doi: 10.3319/TAO.2008.19.6.705(PT)

The following questions should be addressed:

Which formulas and parameters are used, in particular for bottom friction (Manning coefficient)? The bottom friction has an impact on the simulated tsunami amplitude at the coast.

We have added the description of Manning coefficient. [Page 6, lines 9-10] Nonlinear shallow water equation for Cartesian coordinate is used:

$$\frac{\partial \eta}{\partial t} + \frac{\partial P}{\partial x} + \frac{\partial Q}{\partial y} = -\frac{\partial h}{\partial t}$$

$$\frac{\partial P}{\partial t} + \frac{\partial}{\partial x} \left\{ \frac{P^2}{H} \right\} + \frac{\partial}{\partial y} \left\{ \frac{PQ}{H} \right\} + gH \frac{\partial \eta}{\partial x} + F_x = 0$$

$$\frac{\partial Q}{\partial t} + \frac{\partial}{\partial x} \left\{ \frac{PQ}{H} \right\} + \frac{\partial}{\partial y} \left\{ \frac{Q^2}{H} \right\} + gH \frac{\partial \eta}{\partial y} + F_y = 0$$

 $\eta$  is the free-surface displacement. *P* and *Q* are the horizontal volume discharges. *g* is gravity. *h* is the still water depth. *H* is the total water depth,  $H = \eta + h$ . *Fx* and *Fy* are the bottom frictions.

$$F_x = \frac{gn^2}{H^{7/3}} P(P^2 + Q^2)^{1/2}$$
$$F_y = \frac{gn^2}{H^{7/3}} Q(P^2 + Q^2)^{1/2}$$

2

-

n is Manning's roughness coefficient. In this study, Manning coefficient is 0.013, which

represents a smooth surface (Wu, et al., 2008; Wang 2009).

**Reference:**

- Wang, X. (2009). User manual for COMCOT version 1.7 (first draft). Cornel University, 65.
- Wu, T. R., Chen, P. F., Tsai, W. T., and Chen, G. Y.: Numerical Study on Tsunamis Excited by 2006 Pingtung Earthquake Doublet, Terr. Atmos. Ocean. Sci., 19, 705-715, 2008. doi: 10.3319/TAO.2008.19.6.705(PT)

Which bathymetry and topography data is used? Free GEBCO and SRTM? We have added it. [Page 6, lines 10-11]

The resolution of 1 minute for the inner mesh is quite rough for simulations that should give estimates of the tsunami amplitude at the coast. Our experience from hind casts of real events suggests that at the coast line, the horizontal resolution should be 500m (edge length in an unstructured triangular grid) or better. This should be transferable, as COMCOT also is a model with first order spatial discretization.

The Figure 1 presents the time series by uniform slip distribution at station 25 in different resolution of topography. The time series are similar. For resolution, 1 minute is better than 2 minute and for time spent, 1 minute is less than 30 arc-sec. Therefore, to consider the resolution of simulation and time spent, the resolution of 1 minute was applied.

Fig. 1 The time series by uniform slip distribution at station 25 in different resolution of topography. Blue line is 2 minute, red line is 1 minute and yellow line is 30 arc-sec.

Where are the tide gauges located? See also point 14, references. On the one hand, the exact location is not really important, because the study could be performed with virtual sensor locations or coastal forecast points, but

- to reproduce the results, the locations of the (real or virtual) gauges are needed,
- for hind casts of real events, the location and measurements from real tide gauges are needed,

- the simulation of the tsunami wave form at a tide gauge that is located e.g., inside a harbor or narrow bight is very sensitive to errors in the representation of bathymetry and topography (1min resolution for sure is too coarse!) and to the choice of the roughness parameter (wave reflections).
- The comparison in fig. 6 may be spoiled by different gauge locations. Distance to the source is not the only parameter, as it is also stated in the paper, too (e.g., page 7 line 23-24).

We have added location information and removed fitting line [Page 6, lines 13-18; Pages 20-21]

| Table 1. The tide gauge locations in this study. |               |          |         |  |  |  |
|--------------------------------------------------|---------------|----------|---------|--|--|--|
| No.                                              | Station       | Lon      | Lat     |  |  |  |
| 1                                                | Linshanbi     | 121.5106 | 25.2844 |  |  |  |
| 2                                                | Danshuei      | 121.4019 | 25.1844 |  |  |  |
| 3                                                | Jhuwei        | 121.2353 | 25.1200 |  |  |  |
| 4                                                | Hsinchu       | 120.9122 | 24.8503 |  |  |  |
| 5                                                | Waipu         | 120.7717 | 24.6514 |  |  |  |
| 6                                                | Taichung Port | 120.5250 | 24.2917 |  |  |  |
| 7                                                | Fanyuan       | 120.2972 | 23.9147 |  |  |  |
| 8                                                | Bozihliao     | 120.1417 | 23.6250 |  |  |  |
| 9                                                | Penghu        | 119.5669 | 23.5636 |  |  |  |
| 10                                               | Dongshih      | 120.1417 | 23.4417 |  |  |  |
| 11                                               | Jiangjyun     | 120.1000 | 23.2181 |  |  |  |
| 12                                               | Anping        | 120.1583 | 22.9750 |  |  |  |
| 13                                               | Yongan        | 120.1917 | 22.8083 |  |  |  |
| 14                                               | Kaohsiung     | 120.2883 | 22.6144 |  |  |  |
| 15                                               | Donggang      | 120.4417 | 22.4583 |  |  |  |
| 16                                               | Siaoliouciou  | 120.3750 | 22.3583 |  |  |  |
| 17                                               | Jiahe         | 120.6083 | 22.3250 |  |  |  |
| 18                                               | Syunguangzuei | 120.6917 | 21.9917 |  |  |  |
| 19                                               | Houbihu       | 120.7583 | 21.9417 |  |  |  |
| 20                                               | Lanyu         | 121.4917 | 22.0583 |  |  |  |
| 21                                               | Dawu          | 120.8972 | 22.3375 |  |  |  |
| 22                                               | Lyudao        | 121.4647 | 22.6622 |  |  |  |
| 23                                               | Fugang        | 121.1917 | 22.7917 |  |  |  |

We list the location of the gauges in the Table 1.

| 24 | Chenggong | 121.3767 | 23.0889 |
|----|-----------|----------|---------|
| 25 | Shihti    | 121.5250 | 23.4917 |
| 26 | Hualien   | 121.6231 | 23.9803 |
| 27 | Suao      | 121.8686 | 24.5856 |
| 28 | Gengfang  | 121.8619 | 24.9072 |
| 29 | Longdong  | 121.9417 | 25.1250 |
| 30 | Keelung   | 121.7417 | 25.1750 |

The fittings of Fig. 6 just give a rough relationship between wave height and distance for the tsunami source which is perpendicular the coast line. Of course, the distance is not the only parameter for wave height attenuation. We agree to remove the fitting lines.

11. Are mathematical formulae, symbols, abbreviations and units correctly defined and used? If the formulae, symbols or abbreviations are numerous, are there tables or appendixes listing them?

Equation (1): W for width, L for length: It's obvious, but nevertheless should be added in the text above. Which value for  $\mu$  is assumed when estimating Mw? And as a nonseismologist, I would like to ask if the estimate of D = 8.25m is really obvious? Section 2.2: Not my field of expertise at all.

We have done it. [Page 3, lines 21-23]

We will add the definition of symbols (W and L) in the text.  $\mu$  usually sets 30GPa and it assumes that crust is elastically uniform. The estimation of slip and Mw is from fault geometry and parameter assuming as  $\mu$ .

We analyzed the relation between Mw and average slip (D) in Fig 2. The public finite fault slip models of global slip earthquakes are from the website (http://equake-rc.info/SRCMOD/). This figure appears the trend between Mw and average slip and its boundary. For Mw8.15, the range could be 200~1000 cm. It explains that our estimation, which follows the trend and in the possible boundary, is reasonable.

Fig 2. Mw of real events and their average slips with 2 standard deviation (http://equake-rc.info/SRCMOD/). Open circles represent the inverse slip results in each study. Solid circles represent the mean slip of each study for same event.

- ChiChi (1999): Ma et al. (2000); Chi et al. (2001); Zeng and Chen (2001); Wu et al. (2001); Zhang et al. (2004)
- Tohoku (2011): Ammon et al. (2011); Ide et al. (2011); Lay et al. (2011); Shao et al. (2011); Yagi and Fukahata (2011); Yamazaki et al. (2011); Wei et al. (2012)
- Maule (2010): Delouis et al. (2010); Hayes (2010); Shao et al. (2010); Sladen (2010); Luttrell et al. (2011)
- Sumatra (2004): Ammon et al. (2005); Ji (2005); Rhie et al. (2007)
- Sumatra (2012): Hayes (2012); Shao et al. (2012); Wei (2012); Yue et al. (2012)
- Tokachi-Oki (2003): Yamanaka and Kikuchi (2003); Koketsu et al. (2004); Tanioka et al. (2004); Yagi (2004)
- Tocopilla (2007): Ji (2007); Sladen (2007); Zeng et al. (2007); Béjar-Pizarro et al. (2010); Motagh et al. (2010)

**Reference:**

- Ammon, C. J., J. Chen, H.-K. Thio, D. Robinson, S. Ni, V. Hjorleifsdottir, H. Kanamori, T. Lay, S. Das, D. Helmberger, G. Ichinose, J. Polet, and D. Wald. (2005). Rupture process of the great 2004 Sumatra-Andaman earthquake, Science, 308, 1133-1139.
- Ammon, C. J., T. Lay, H. Kanamori, and M. Cleveland (2011) A rupture model of the 2011 off the Pacific coast of Tohoku Earthquake, Earth Planets Space, 63, 693– 696.
- Bejar-Pizzaro M., Carrizo D., Socquet A., Armijo R., (2010) Asperities, barriers and

transition zone in the North Chile seismic gap: State of the art after the 2007 Mw 7.7 Tocopilla earthquake inferred by GPS and InSAR data, Geoph. Journ. Int., GJI-S-09-0648, doi: 10.1111/j.1365-246X.2010.04748.x

- Chi, W. C., D. Dreger, and A. Kaverina. 2001. Finite-source modeling of the 1999 Taiwan (Chi-Chi) earthquake derived from a dense strong-motion network. Bull. Seis. Soc. Am 91 (5):1144-1157.
- Delouis B., J. M. Nocquet, M. Vallée (2010). Slip distribution of the February 27, 2010 Mw = 8.8 Maule Earthquake, central Chile, from static and high-rate GPS, InSAR, and broadband teleseismic data, Geophys. Res. Lett., 37, L17305, doi:10.1029/2010GL043899.
- Hayes G., (NEIC, Maule 2010) Updated Result of the Feb 27, 2010 Mw 8.8 Maule, Chile Earthquake, http://earthquake.usgs.gov/earthquakes/eqinthenews/2010/us2010tfan/finite\_fault .php,last accessed August 19, 2013.
- Hayes G., (NEIC, Sumatra 2012) Preliminary Result of the Apr 11, 2012 Mw 8.6 Earthquake Off the West Coast of Northern Sumatra, http://earthquake.usgs.gov/earthquakes/eqinthenews/2012/usc000905e/finite\_fau lt.php,last accessed August 19, 2013.
- Ide S., A. Baltay, and G. C. Beroza (2011). Shallow Dynamic Overshoot and Energetic Deep Rupture in the 2011 Mw 9.0 Tohoku-Oki Earthquake, 332, 1426-1429, DOI: 10.1126/science.1207020
- Ji, C. (2005). Preliminary Rupture Model for the December 26, 2004 earthquake, off the west coast of northern Sumatra, magnitude 9.1, http://neic.usgs.gov/neis/eq\_depot/2004/eq\_041226/neic\_slav\_ff.html.
- Ji C. (UCSB, Tocopilla 2007) Preliminary Result of the Nov 14, 2007 Mw 7.81 ANTOFAGASTA, CHILE Earthquake, http://www.geol.ucsb.edu/faculty/ji/big\_earthquakes/2007/11/anto/anto.html, last accessed August 11, 2013.
- Koketsu, K., K. Hikima, S. Miyazaki, and S. Ide. (2004). Joint inversion of strong motion and geodetic data for the source process of the 2003 Tokachi-oki, Hokkaido, earthquake. Earth Planets and Space 56 (3):329-334.
- Lay T., C. J. Ammon, H. Kanamori, L. Xue, and M. J. Kim (2011). Possible large neartrench slip during the 2011 Mw 9.0 off the Pacific coast of Tohoku Earthquake. Earth Planets Space. 63, 687–692.
- Luttrell, K. M., Tong, X., Sandwell, D. T., Brooks, B. A., and Bevis, M. G. (2011). Estimates of stress drop and crustal tectonic stress from the 27 February 2010 Maule, Chile, earthquake: Implications for fault strength. Journal of Geophysical Research: Solid Earth (1978–2012), 116(B11).

- Ma, K. F., T. R. A. Song, S. J. Lee, and H. I. Wu. (2000). Spatial slip distribution of the September 20, 1999, Chi-Chi, Taiwan, earthquake (M(W)7.6) - Inverted from teleseismic data. Geophys. Res. Lett. 27 (20):3417-3420.
- Motag, M., B. Schurr, J. Anderssohn, B. Cailleau, T. R. Walter, R. Wang, J.-P. Villotte, (2010) Subduction earthquake deformation associated with 14 November 2007, Mw 7.8 Tocopilla earthquake in Chile: Results from InSAR and aftershocks, Tectonophysics 490, 60–68
- Rhie, J., D. Dreger, R. Burgmann, and B. Romanowicz. (2007). Slip of the 2004 Sumatra–Andaman Earthquake from joint inversion of long-period global seismic waveforms and GPS static Offsets, Bull. Seismo. Soc. Am., 97(1A): S115–S127.
- Shao, G., X. Li and C. Ji. (UCSB, sumatra 2012). Preliminary Result of the Apr 11, 2012 Mw 8.64 sumatra Earthquake, http://www.geol.ucsb.edu/faculty/ji/big\_earthquakes/2012/04/10/sumatra.html,la st accessed August 19, 2013.
- Shao, G., X. Li, C. Ji. and T. Maeda (2011). Focal mechanism and slip history of 2011 Mw 9.1 off the Pacific coast of Tohoku earthquake, constrained with teleseismic body and surface waves, Earth Planets Space, 63 (7), 559-564.
- Shao, G., X. Li, Q. Liu, X. Zhao, T. Yano and C. Ji(UCSB, Maule 2010).Preliminary slip model of the Feb 27, 2010 Mw 8.9 Maule, Chile Earthquake, http://www.geol.ucsb.edu/faculty/ji/big\_earthquakes/2010/02/27/chile\_2\_27.html ,last accessed September 24,2013.
- Sladen A. (Caltech, Tocopilla 2007). Preliminary Result 11/14/2007 (Mw 7.7), Tocopilla Earthquake, Chile. Source Models of Large Earthquakes. http://www.tectonics.caltech.edu/slip\_history/2007\_tocopilla/tocopilla.html, last accessed July 1, 2013.
- Sladen A. (Caltech, Maule 2010). Preliminary Result, 02/27/2010 (Mw 8.8), Chile.SourceModelsofLargeEarthquakes.http://www.tectonics.caltech.edu/slip\_history/2010\_chile/index.html.
- Tanioka, Y., K. Hirata, R. Hino, and T. Kanazawa. (2004). Slip distribution of the 2003 Tokachi-oki earthquake estimated from tsunami waveform inversion. Earth Planets and Space 56 (3):373-376.
- Wei S. (Caltech, Sumatra 2012). April/11/2012 (Mw 8.6), Sumatra. Source Models of Large Earthquakes. http://www.tectonics.caltech.edu/slip\_history/2012\_Sumatra/index.html, last accessed July 1, 2013.
- Wei, S. J., R.W. Graves, D. Helmberger, J.P. Avouac and J.L. Jiang (2012) Sources of shaking and flooding during the Tohoku-Oki earthquake: A mixture of rupture styles, Earth and Planetary Science Letters, 333-334, 91-100.

- Wu, C. J., M. Takeo, and S. Ide. (2001). Source process of the Chi-Chi earthquake: A joint inversion of strong motion data and global positioning system data with a multifault model. Bull. Seis. Soc. Am 91 (5):1128-1143.
- Yagi, Y. (2004). Source rupture process of the 2003 Tokachi-oki earthquake determined by joint inversion of teleseismic body wave and strong ground motion data, Earth Planets Space, 56, 311–316.
- Yagi, Y. and Fukahata, Y., (2011). Rupture process of the 2011 Tohoku-oki earthquake and absolute elastic strain release, Geophys. Res. Lett, 38, L19307, doi:10.1029/2011GL048701.
- Yamanaka, Y., and M. Kikuchi. (2003). Source process of the recurrent Tokachi-oki earthquake on September 26, 2003, inferred from teleseismic body waves. Earth Planets and Space 55 (12):E21-E24.
- Yamazaki, Y., T. Lay, K. F. Cheung, H. Yue, and H. Kanamori (2011). Modeling nearfield tsunami observations to improve finite-fault slip models for the 11 March 2011 Tohoku earthquake, Geophys. Res. Lett., 38, L00G15, doi:10.1029/2011GL049130.
- Yue, H, T. Lay and K. D. Koper (2012), En Echelon andOrthogonal Fault Ruptures of the 11 April 2012 Great Intraplate Earthquakes. Nature, 490, 245-249, doi:10.1038/nature11492.
- Zeng, Y. H., and C. H. Chen. (2001). Fault rupture process of the 20 September 1999 Chi-Chi, Taiwan, earthquake. Bull. Seis. Soc. Am 91 (5):1088-1098.
- Zeng, Y., G.Hayes and C. Ji (2007; USGS, Online Model). Preliminary Result of the Nov 14, 2007 Mw 7.7 Antofagasto, Chile Earthquake, http://earthquake.usgs.gov/earthquakes/eqinthenews/2007/us2007jsat/finite\_fault .php, last accessed August 20, 2013.
- Zhang, W., T. Iwata, K. Irikura, A. Pitarka, and H. Sekiguchi (2004), Dynamic rupture process of the 1999 Chi-Chi, Taiwan, earthquake, Geophys. Res. Lett., 31, L10605, doi:10.1029/2004GL019827.

12. Is the size, quality and readability of each figure adequate to the type and quantity of data presented?

Figure 4: change y-axis label to "Wave amplitude"

Figure 6: I would keep this figure, but skip the explicit linear fitting. It pretends an accuracy that cannot be obtained.

We have done it. [Pages 18, 20]

14. Are the number and quality of the references appropriate?

A citation for the tide gauge locations or at least a list of coordinates would be handy. The Taiwanese tide gauges are not available at http://www.ioc-sealevelmonitoring.org or http://www.psmsl.org/ (Taipei until 1995, Kaohsiung until 1996), and I could not find a link to the gauges at the website of the Taiwanese Central Weather Bureau (CWB) http://www.cwb.gov.tw This private/commercial site was the best information I could find: https://www.tide-forecast.com/locations/Hualien-City . Still, no exact location, but the "Detailed Map" gives at least an idea that this station is located inside the harbour. In total, 9 Taiwanese stations are available here. I am missing a short overview of historical tsunamis in Taiwan, but the last local tsunami occurred in 1867, and it might be difficult to find scientific papers to cite. e.g., see http://scweb.cwb.gov.tw/NewsContent.aspx?ItemId=37&CId=199&loc=en However, I found the following paper - no tsunami, but a report on the uplift of the tide

gauge due to the earthquake. Maybe, this paper provides a helpful hindcast, too: COMCOT should not show a strong tsunami. Chung-Liang Lo, Emmy Tsui-Yu Chang, and Benjamin Fong Chao. Relocating the historical 1951 Hualien earthquake in eastern Taiwan based on tide gauge record. Geophys. J. Int. (2013) 192, 854–860. doi: 10.1093/gji/ggs058

We have added the information of location. [Page 6, lines 13-18]

The website of Taiwanese Central Weather Bureau (CWB) presents the location of tidestations(http://e-service.cwb.gov.tw/HistoryDataQuery/index.jsp andhttp://www.cwb.gov.tw/V7e/climate/marine\_stat/tide.htm).

Lo et al., (2013) investigated the historical 1951 Hualien earthquake sequence. The magnitude of three earthquakes are smaller than our scenario estimation and the focal mechanisms are different from our fault model so that it is not applicable to be compared with our study. This maybe be considered another tsunami earthquake.

15. Are the references accessible by fellow scientists? Yes, but please add doi numbers.

We have done it. [Pages 11-15]

Reviewer #2:

Page 2, lines 3-5. If the earthquakes associated to the historic tsunamis mentioned in the text have any magnitude estimation, please provide the value and include the reference. For instance, the 1867 tsunami, magnitude?

We have done it. [Page 2, lines 3-4]

The 1867 Keelung earthquake was inferred approximately Mw 7.0 (Tsai 1985; Ma and Lee 1997; Cheng et al., 2016; Yu et al., 2016).

**Reference:**

- Cheng, S. N., Shaw, C. F., and Yeh, Y. T. (2016). Reconstructing the 1867 Keelung Earthquake and Tsunami Based on Historical Documents. Terr. Atmos. Ocean. Sci., 27(3). doi: 10.3319/TAO.2016.03.18.01(TEM)
- Ma, K. F., and Lee, M. F. (1997). Simulation of historical tsunamis in the Taiwan region. Terr. Atmos. Ocean. Sci., 8(1), 13-30. doi: 10.3319/TAO.1997.8.1.13(T)
- Tsai, Y. B. (1985). A study of disastrous earthquakes in Taiwan, 1683–1895. Bull. Inst. Earth Sci. Acad. Sin, 5, 1-44.
- Yu, N.-T., Yen, J.-Y., Chen, W.-S., Yen, I. C., and Liu, J.-H.: Geological records of western Pacific tsunamis in northern Taiwan: AD 1867 and earlier event deposits, Mar. Geol., 372, 1-16, 2016. doi:10.1016/j.margeo.2015.11.010

Page 2, lines 13-15. When comparing PTHA and PSHA, authors mentioned in the text that PSHA works with ground-motion parameters. So, can you complete the idea by specifying that PTHA works with tsunami wave amplitudes, or some other wave measurements? If there is any reference, please include it.

We have done it. [Page 2, lines 14-19]

Geist and Parsons (2006) mentions that the tsunami wave amplitudes follow a definable frequency-size distribution over a sufficiently long amount of time at a given coastal region (Soloviev, 1969; Houston et al., 1977; Horikawa and Shuto, 1983; Burroughs and Tebbens, 2005). This method is of great use in establishing tsunami probability for regions if there is an extensive catalog of observed tsunami wave heights (Geist and Parsons, 2006). The other approach is numerical simulation (Geist, 2002; Geist and Parsons, 2006; Geist and Parsons, 2009) which applies the stochastic slip model to estimate the tsunami amplitudes probability as this study.

**Reference:**

Burroughs, S.M., Tebbens, S.F. (2005). Power law scaling and probabilistic forecasting of tsunami runup heights. Pure Appl. Geophys. 162, 331–342

- Geist, E.L., (2002). Complex earthquake rupture and local tsunamis. J. Geophys. Res. 107. doi:10.1029/2000JB000139.
- Geist, E. L., and Parsons, T. (2006). Probabilistic analysis of tsunami hazards. Natural Hazards, 37(3), 277-314. doi 10.1007/s11069-005-4646-z
- Geist, E. L., and Parsons, T. (2009). Assessment of source probabilities for potential tsunamis affecting the US Atlantic coast. Marine Geology, 264(1), 98-108.
- Horikawa, K. and Shuto, N. (1983). Tsunami disasters and protection measures in Japan, In: K. Iida and T. Iwasaki (eds), Tsunamis-Their Science and Engineering, Terra Scientific Publishing Company, pp. 9–22.
- Houston, J. R., Carver, R. D. and Markle, D. G. (1977). Tsunami-wave elevation frequency of occurrence for the Hawaiian Islands. Technical Report H-77-16, U.S. Army Engineer Waterways Experiment Station, Vicksburg, MS, 66 pp.
- Soloviev, S. L. (1969). Recurrence of tsunamis in the Pacific. In: W. M. Adams (ed.), Tsunamis in the Pacific Ocean, East-West Center Press, pp. 149–163.

Page 3. Line 2. Please, provide the reference for the magnitude range, Mw 7.5-8.7. We have done it. [Page 3, line 5]

**Reference:**

Hsu, Y. J., Ando, M., Yu, S. B., and Simons, M. (2012). The potential for a great earthquake along the southernmost Ryukyu subduction zone. Geophysical Research Letters, 39(14).

Page 3. Line 12. About the fault geometry setting. Which is the source depth of the top (or bottom) of the fault plane? I think it has not been specified yet in the text.

We have done it. [Page 3, lines 14-15]

The fault geometry setting refers to Hsu et al. (2012) and fault model extends from the Ryukyu Trench to a depth of 13 km.

Page 3. Line 15, please complete to "...in dip slip faults". Thank you. We have done it. [Page 3, line 18]

Page 3. Eq. (1), please, specify what is L, and W. We have done it. [Page 3, line 21]

Page 3. Line 18. I suggest to change "constant" by "parameter". Strictly speaking, in elastic heterogeneous media, the Lamè parameters (lambda and mu) vary in space. We have done it. [Page 3, line 22]

Page 3. In Eq. (2). Which is the value assumed for mu? We have done it. [Page 3, lines 22-23]

Page 3. Section 2.1. When the authors compute the earthquake magnitude, average slip and fault area. Did the authors compare (or contrast) these values with any magnitude/fault-size scaling relationship for subduction earthquakes? It could be interesting to compare these values with any magnitude/size scaling relationship for subduction zones.

We analyzed the relation between Mw and average slip (D) in Fig 1. The public finite fault slip models of global slip earthquakes are from the website (http://equake-rc.info/SRCMOD/). This figure appears the trend between Mw and average slip and its boundary. For Mw8.15, the range could be  $200 \sim 1000$  cm. It explains that our estimation, which follows the trend and in the possible boundary, is reasonable.

Fig 1. Mw of real events and their average slips with 2 standard deviation (http://equake-rc.info/SRCMOD/). Open circles represent the inverse slip results in each study. Solid circles represent the mean slip of each study for same event.

- ChiChi (1999): Ma et al. (2000); Chi et al. (2001); Zeng and Chen (2001); Wu et al. (2001); Zhang et al. (2004)
- Tohoku (2011): Ammon et al. (2011); Ide et al. (2011); Lay et al. (2011); Shao et al. (2011); Yagi and Fukahata (2011); Yamazaki et al. (2011); Wei et al. (2012)
- Maule (2010): Delouis et al. (2010); Hayes (2010); Shao et al. (2010); Sladen (2010); Luttrell et al. (2011)
- Sumatra (2004): Ammon et al. (2005); Ji (2005); Rhie et al. (2007)

Sumatra (2012): Hayes (2012); Shao et al. (2012); Wei (2012); Yue et al. (2012)

- Tokachi-Oki (2003): Yamanaka and Kikuchi (2003); Koketsu et al. (2004); Tanioka et al. (2004); Yagi (2004)
- Tocopilla (2007): Ji (2007); Sladen (2007); Zeng et al. (2007); Béjar-Pizarro et al. (2010); Motagh et al. (2010)

**Reference:**

- Ammon, C. J., J. Chen, H.-K. Thio, D. Robinson, S. Ni, V. Hjorleifsdottir, H. Kanamori, T. Lay, S. Das, D. Helmberger, G. Ichinose, J. Polet, and D. Wald. (2005). Rupture process of the great 2004 Sumatra-Andaman earthquake, Science, 308, 1133-1139.
- Ammon, C. J., T. Lay, H. Kanamori, and M. Cleveland (2011) A rupture model of the 2011 off the Pacific coast of Tohoku Earthquake, Earth Planets Space, 63, 693– 696.
- Bejar-Pizzaro M., Carrizo D., Socquet A., Armijo R., (2010) Asperities, barriers and transition zone in the North Chile seismic gap: State of the art after the 2007 Mw 7.7 Tocopilla earthquake inferred by GPS and InSAR data, Geoph. Journ. Int., GJI-S-09-0648, doi: 10.1111/j.1365-246X.2010.04748.x
- Chi, W. C., D. Dreger, and A. Kaverina. 2001. Finite-source modeling of the 1999 Taiwan (Chi-Chi) earthquake derived from a dense strong-motion network. Bull. Seis. Soc. Am 91 (5):1144-1157.
- Delouis B., J. M. Nocquet, M. Vallée (2010). Slip distribution of the February 27, 2010 Mw = 8.8 Maule Earthquake, central Chile, from static and high-rate GPS, InSAR, and broadband teleseismic data, Geophys. Res. Lett., 37, L17305, doi:10.1029/2010GL043899.
- Hayes G., (NEIC, Maule 2010) Updated Result of the Feb 27, 2010 Mw 8.8 Maule, Chile Earthquake, http://earthquake.usgs.gov/earthquakes/eqinthenews/2010/us2010tfan/finite\_fault .php,last accessed August 19, 2013.
- Hayes G., (NEIC, Sumatra 2012) Preliminary Result of the Apr 11, 2012 Mw 8.6 Earthquake Off the West Coast of Northern Sumatra, http://earthquake.usgs.gov/earthquakes/eqinthenews/2012/usc000905e/finite\_fau lt.php,last accessed August 19, 2013.
- Ide S., A. Baltay, and G. C. Beroza (2011). Shallow Dynamic Overshoot and Energetic Deep Rupture in the 2011 Mw 9.0 Tohoku-Oki Earthquake, 332, 1426-1429, DOI: 10.1126/science.1207020
- Ji, C. (2005). Preliminary Rupture Model for the December 26, 2004 earthquake, off the west coast of northern Sumatra, magnitude 9.1, http://neic.usgs.gov/neis/eq\_depot/2004/eq\_041226/neic\_slav\_ff.html.

- Ji C. (UCSB, Tocopilla 2007) Preliminary Result of the Nov 14, 2007 Mw 7.81 ANTOFAGASTA, CHILE Earthquake, http://www.geol.ucsb.edu/faculty/ji/big\_earthquakes/2007/11/anto/anto.html, last accessed August 11, 2013.
- Koketsu, K., K. Hikima, S. Miyazaki, and S. Ide. (2004). Joint inversion of strong motion and geodetic data for the source process of the 2003 Tokachi-oki, Hokkaido, earthquake. Earth Planets and Space 56 (3):329-334.
- Lay T., C. J. Ammon, H. Kanamori, L. Xue, and M. J. Kim (2011). Possible large neartrench slip during the 2011 Mw 9.0 off the Pacific coast of Tohoku Earthquake. Earth Planets Space. 63, 687–692.
- Luttrell, K. M., Tong, X., Sandwell, D. T., Brooks, B. A., and Bevis, M. G. (2011). Estimates of stress drop and crustal tectonic stress from the 27 February 2010 Maule, Chile, earthquake: Implications for fault strength. Journal of Geophysical Research: Solid Earth (1978–2012), 116(B11).
- Ma, K. F., T. R. A. Song, S. J. Lee, and H. I. Wu. (2000). Spatial slip distribution of the September 20, 1999, Chi-Chi, Taiwan, earthquake (M(W)7.6) - Inverted from teleseismic data. Geophys. Res. Lett. 27 (20):3417-3420.
- Motag, M., B. Schurr, J. Anderssohn, B. Cailleau, T. R. Walter, R. Wang, J.-P. Villotte, (2010) Subduction earthquake deformation associated with 14 November 2007, Mw 7.8 Tocopilla earthquake in Chile: Results from InSAR and aftershocks, Tectonophysics 490, 60–68
- Rhie, J., D. Dreger, R. Burgmann, and B. Romanowicz. (2007). Slip of the 2004 Sumatra–Andaman Earthquake from joint inversion of long-period global seismic waveforms and GPS static Offsets, Bull. Seismo. Soc. Am., 97(1A): S115–S127.
- Shao, G., X. Li and C. Ji. (UCSB, sumatra 2012). Preliminary Result of the Apr 11, 2012 Mw 8.64 sumatra Earthquake, http://www.geol.ucsb.edu/faculty/ji/big\_earthquakes/2012/04/10/sumatra.html,la st accessed August 19, 2013.
- Shao, G., X. Li, C. Ji. and T. Maeda (2011). Focal mechanism and slip history of 2011 Mw 9.1 off the Pacific coast of Tohoku earthquake, constrained with teleseismic body and surface waves, Earth Planets Space, 63 (7), 559-564.
- Shao, G., X. Li, Q. Liu, X. Zhao, T. Yano and C. Ji(UCSB, Maule 2010).Preliminary slip model of the Feb 27, 2010 Mw 8.9 Maule, Chile Earthquake, http://www.geol.ucsb.edu/faculty/ji/big\_earthquakes/2010/02/27/chile\_2\_27.html ,last accessed September 24,2013.
- Sladen A. (Caltech, Tocopilla 2007). Preliminary Result 11/14/2007 (Mw 7.7), Tocopilla Earthquake, Chile. Source Models of Large Earthquakes. http://www.tectonics.caltech.edu/slip\_history/2007\_tocopilla/tocopilla.html, last

accessed July 1, 2013.

- Sladen A. (Caltech, Maule 2010). Preliminary Result, 02/27/2010 (Mw 8.8), Chile. Source Models of Large Earthquakes. http://www.tectonics.caltech.edu/slip history/2010 chile/index.html.
- Tanioka, Y., K. Hirata, R. Hino, and T. Kanazawa. (2004). Slip distribution of the 2003 Tokachi-oki earthquake estimated from tsunami waveform inversion. Earth Planets and Space 56 (3):373-376.
- Wei S. (Caltech, Sumatra 2012). April/11/2012 (Mw 8.6), Sumatra. Source Models of Large Earthquakes. http://www.tectonics.caltech.edu/slip\_history/2012\_Sumatra/index.html, last accessed July 1, 2013.
- Wei, S. J., R.W. Graves, D. Helmberger, J.P. Avouac and J.L. Jiang (2012) Sources of shaking and flooding during the Tohoku-Oki earthquake: A mixture of rupture styles, Earth and Planetary Science Letters, 333-334, 91-100.
- Wu, C. J., M. Takeo, and S. Ide. (2001). Source process of the Chi-Chi earthquake: A joint inversion of strong motion data and global positioning system data with a multifault model. Bull. Seis. Soc. Am 91 (5):1128-1143.
- Yagi, Y. (2004). Source rupture process of the 2003 Tokachi-oki earthquake determined by joint inversion of teleseismic body wave and strong ground motion data, Earth Planets Space, 56, 311–316.
- Yagi, Y. and Fukahata, Y., (2011). Rupture process of the 2011 Tohoku-oki earthquake and absolute elastic strain release, Geophys. Res. Lett, 38, L19307, doi:10.1029/2011GL048701.
- Yamanaka, Y., and M. Kikuchi. (2003). Source process of the recurrent Tokachi-oki earthquake on September 26, 2003, inferred from teleseismic body waves. Earth Planets and Space 55 (12):E21-E24.
- Yamazaki, Y., T. Lay, K. F. Cheung, H. Yue, and H. Kanamori (2011). Modeling nearfield tsunami observations to improve finite-fault slip models for the 11 March 2011 Tohoku earthquake, Geophys. Res. Lett., 38, L00G15, doi:10.1029/2011GL049130.
- Yue, H, T. Lay and K. D. Koper (2012), En Echelon andOrthogonal Fault Ruptures of the 11 April 2012 Great Intraplate Earthquakes. Nature, 490, 245-249, doi:10.1038/nature11492.
- Zeng, Y. H., and C. H. Chen. (2001). Fault rupture process of the 20 September 1999 Chi-Chi, Taiwan, earthquake. Bull. Seis. Soc. Am 91 (5):1088-1098.
- Zeng, Y., G.Hayes and C. Ji (2007; USGS, Online Model). Preliminary Result of the Nov 14, 2007 Mw 7.7 Antofagasto, Chile Earthquake, http://earthquake.usgs.gov/earthquakes/eqinthenews/2007/us2007jsat/finite\_fault

.php, last accessed August 20, 2013.

Zhang, W., T. Iwata, K. Irikura, A. Pitarka, and H. Sekiguchi (2004), Dynamic rupture process of the 1999 Chi-Chi, Taiwan, earthquake, Geophys. Res. Lett., 31, L10605, doi:10.1029/2004GL019827.

Page 3, line 25. For completeness purposes, please provide the scalar seismic moment, M0 for the corresponding Mw 8.15. We have done it. [Page 4, line 2]

Page 4. Please clarify or complete the sentence in line 8, because there is a dot at the end of the sentence, so it is not clear what Eq. (4) means or represents. The 2D Fourier spectrum amplitude of what?

We have done it. [Page 4, lines 9-11]

Eq. (4) illustrate that the spectrum of static slip distribution in wavenumber domain is following  $k^{-2}$  decay. In Eq. (4),  $D_{xy}$  is slip distribution and its spectrum is proportional to  $k^{-2}$ . And rews (1980) derived the  $k^{-2}$  from the relationship of slip and stress change.

Page 4. Line 10. Please, to be consistent with the notation in Eq. (4), please clarify the meaning of "F", or, change F by Fs,t which represents the 2D discrete Fourier transform of Dx,y. Also, for completeness purposes, specify that Dx,y is the slip distribution over a 2D lattice, for instance.

Thank you. We have done it. [Page 4, line 13]

Page 4. In line 10, please complete, "...wave number.", by "...radial wavenumber." Thank you. We have done it. [Page 4, line 13]

Page 4. Line 13, please correct "corner frequency" by "corner radial wavenumber", because kc is not a frequency.

Thank you. We have done it. [Page 4, line 16; Page 5, line 7 and Page 15, line 13]

Page 4. Line 14. What happen with the phase beyond kc? Please, clarify. Or, the last sentence "Within the kc,....(Geist, 2002)." could be deleted because authors are describing the overall characteristics of the slip and not describing the details of how the random slip is generated numerically in the practice.

We have removed this sentence. Beyond the corner radial wavenumber, kc, the slip spectrum decays with  $k^{-2}$ . The generation of random slip is explained in next paragraph, Page 5.

Page 4. Eq. (5). Please, be careful and clear with the mathematical notation. What does F(-1) represent ?. Is it the inverse 2D discrete Fourier transform? Thank you. We have done it. [Page 4, lines 22-23]

Page 4. Line 23. Please, specify that PDF is Probability Density Function, I think it has not been mentioned before in the text. Thank you. We have done it. [Page 4, line 27]

PDF is Probability Density Function.

Page 5. Line 3. Complete the units in the sentence, "...5x5 km...", by "...5x5 km^2...". Thank you. We have done it. [Page 5, line 8; Page 5, line 32; Page 16, line 4]

Page 5. Line 3. Please, clarify that 24x14 are along strike and dip respectively. Thank you. We have done it. [Page 5, line 8]

Page 5. Line 1-4. I will ask the authors to provide some details about how the stochastic slip distribution is generated, and to be clear on the choice of parameters and discuss about the results. Please, read the following comments.

The authors used the values of the Levy PDF suggested by Lavallee et al. (2006), so please clarify in the manuscript that those values were estimated from a stochastic 2D model in the dip slip direction, obtained for the Northridge earthquake. So, why do you use parameters from a shallow crustal earthquake occurred in California to characterize a interplate subduction zone earthquake? Please justify, or discuss.

A: Thank you. We have done it. [Page 5, lines 10-15]

Furthermore, in this study, we do not focus on the values of characteristic for different kinds of faults. Therefore, we decided to simply apply these values which had been published already.

**Reference:**

Davis, T. L. (1994). 1994 Northridge earthquake. Nature, 372, 167.

Notice that according to Lavallee et al (2006) and others, the scaling exponent is (nu+1) so, the Power Spectrum Density of slip is,  $P(k) \sim k^{(-(nu+1))}$ , it implies that the slip spectrum behaves as,  $D(k) \sim k^{(-(nu+1)/2)}$ . The authors generate random variables using the Levy distribution, and imposed  $P(k) \sim k^{(-2)}$  as shown in Fig. 1c, so, the slip in the wavenumber domain behaves as,  $D(k) \sim k^{(-1)}$ , and Figure 1 is ok, but the slip spectrum does not follow the  $k^{(-2)}$  source characteristic discussed at the beginning of

Section 2.2. Please, clarify this point in the text. Also, discuss the effect in the spatial distribution of slip of this choice (falloff as  $k^{(-1)}$  of the slip spectrum amplitude in the wavenumber domain), versus a slip spectrum that falloff as  $k^{(-2)}$ . From the results shown in Fig. 1, authors generated a slip spectrum that decays as  $k^{-1}$  because they imposed the power spectrum density as  $P(k) \sim k^{(-2)}$ , but in the legend they say "This slip spectrum decays with exponent of -2 and...", so, it is an inconsistency for me. Please, be clear on the choice, and the terminology used when generating spatial random fields. Herrero & Bernard (1994), Andrews (1981), and others, used a stochastic slip model with a 2D Fourier spectrum that decays as  $k^2$  which means,  $D(k) \sim k^2(-2)$ . I am not saying the authors are wrong in their choice, it is only that some parts of the text need some clarification, justification of the choice, or discussion about the assumptions done. We are very sorry for the confusion. In general, the spectrum of slip distribution is proportional to k-2 (Herrero and Bernard 1994; Andrews 1980; Tsai 1997). (|D(k)|~k^(nu-1), nu=1) At the beginning of Section 2.2, the Eq. (1) wants to present the spectrum of slip distribution is proportional to k-2. Fig. 1c shows slip spectrum and it consist with k-square. In Lavallee et al (2006), it is formularized by power spectrum density so that there is a disparity of square. We have modified the sentence and Fig. 1c. [Page 4, lines 9-11; Page 15]

**Page 5. Line 3. Why did you set a 5x5 subfault size? Did you test different subfault sizes?**

For  $5x5 \text{ km}^2$ , the resolution of 1 minute (~1.8 km) should be enough to calculate and differentiate the surface deformation.

Page 5. Line 3. Did you assume a constant slip at each subfault? If it is the case, how do you treat the non-smooth slip boundary condition at the boundaries of the fault? Did you apply a taper at all the borders, if not, authors should discuss or justify their treatment?

Page 5. Lines 15-19. Same comment as done in Page 5, line 3, about the assumption of uniform slip at each subfault.

Thank you. We have done it. [Page 5, lines 22-27]

In this study, we do not do any smooth for slip distribution or its boundary. They are complete uniform slip and stochastic process over the fault model. There are two reasons for this application. The first is that we do not have information for where is locked or the location of asperity often repeats in historical event. The second is that there are some studies present the asperity expanding to the boundary of fault model (Ide et al., 2011; Lay et al., 2011; Shao et al., 2011; Yue and Lay 2011). According to these, we do not prefer to apply any extra constraint. If we have more information about

the characteristic of rupture behavior for this region, we would consider giving a constraint.

**Reference:**

- Ide, S., Baltay, A., and Beroza, G. C. (2011). Shallow dynamic overshoot and energetic deep rupture in the 2011 Mw 9.0 Tohoku-Oki earthquake. *Science*, 332(6036), 1426-1429. doi: 10.1126/science.1207020
- Lay, T., Ammon, C. J., Kanamori, H., Xue, L., and Kim, M. J. (2011). Possible large near-trench slip during the 2011 Mw 9.0 off the Pacific coast of Tohoku Earthquake. *Earth, planets and space*, 63(7), 32. doi:10.5047/eps.2011.05.033
- Shao, G., Li, X., Ji, C., and Maeda, T. (2011). Focal mechanism and slip history of the 2011 Mw 9.1 off the Pacific coast of Tohoku Earthquake, constrained with teleseismic body and surface waves. *Earth, planets and space*, 63(7), 9. doi:10.5047/eps.2011.06.028
- Yue, H., and Lay, T. (2011). Inversion of high-rate (1 sps) GPS data for rupture process of the 11 March 2011 Tohoku earthquake (Mw 9.1). *Geophysical Research Letters*, 38(7). doi: 10.1029/2011GL048700

Page 5, line 15. I would suggest to use "computational domain" instead of "...numerical model".

Thank you. We have done it. [Page 5, line 32]

Page 5. Line 15. Complete the units in 5x5 km2. Thank you. We have done it. [Page 5, line 32].

Page 5, lines 21-25. Why do you use 4 min and 1 min for the nested grids? Did you test a different grid size? Which bathymetry/topography is used in the numerical simulation of the tsunami? Please include a reference. For instance, GEBCO (https://www.gebco.net/) provides a global 30 arc-sec bathymetry, which has a better resolution than the bathymetry used in this work. Please comment on it. Which is the boundary condition set at the coastlines (the boundary between wet and dry domains)?. Do you assume a vertical wall condition, or do you allow inundation? Did you impose any friction, if yes, which one is the Manning's coefficient used in the simulation? Thank you. We have done it. [Page 6, lines 11-12; Page 6, lines 13-18]

NOAA's open data is used. It is free GEBCO and SRTM. The data can be download from: https://maps.ngdc.noaa.gov/viewers/wcs-client/

The Figure 2 presents the time series by uniform slip distribution at station 25 in different resolution of topography. The time series are similar. For resolution, 1 minute

is better than 2 minute and for time spent, 1 minute is less than 30 arc-sec. Therefore, to consider the resolution of simulation and time spent, the resolution of 1 minute was applied. COMCOT is capable of efficiently studying the entire life-span of a tsunami, including its generation, propagation, runup and inundation. COMCOT also supports the nested grid system that the finer grid can be placed on a coarser grid to increase the resolution locally (Wang 2009). In this study, Manning coefficient is 0.013, which represents a smooth surface (Wu, et al., 2008).

---

## Author Response (AR2)

**Re: (nhess-2017-336) Assessment of peak tsunami amplitude associated with a great earthquake occurring along the southernmost Ryukyu subduction zone for Taiwan region** *by* **Yu-Sheng Sun, Po-Fei Chen, Chien-Chih Chen, Ya-Ting Lee, Kuo-Fong Ma, and Tso-Ren Wu**

Dear Prof. Lionello,

Thank you for reviewing this paper. We have made the revision to our manuscript intensively and reply the comments from reviewers carefully for your further consideration on the publication in Natural Hazards and Earth System Sciences (*NHESS*).

The authors highly appreciate the support of publication in *NHESS* from the reviewers and their helpful suggestion as well. We have made substantive modifications according to their suggestion and the **English editing by Springer Nature**. The annotated responses to the reviewers' comments and the details about our changes in the revised version of our manuscript are made accordingly in the files. Attached please also find the electronic files of the revised manuscript for your further consideration of publication in *NHESS*. In the revised version, all modifications were marked in red for your reference. Any problem raised please let me know. Thank you very much.

With Best Regards,
Yu-Sheng Sun

**Nature Research Editing Service Certification**

This document certifies that the manuscript listed below was edited for proper English language, grammar, punctuation, spelling, and overall style by one or more of the highly qualified native English speaking editors at American Journal Experts.

This certificate may be verified at www.aje.com/certificate. This document certifies that the manuscript listed above was edited for proper English language, grammar, punctuation, spelling, and overall style by one or more of the highly qualified native English speaking editors at American Journal Experts. Neither the research content nor the authors' intentions were altered in any way during the editing process. Documents receiving this certification should be English-ready for publication; however, the author has the ability to accept or reject our suggestions and changes. To verify the final AJE edited version, please visit our verification page. If you have any questions or concerns about this edited document, please contact American Journal Experts at support@aje.com.

**Manuscript title:** Assessment of peak tsunami amplitude associated with a great earthquake occurring along the southernmost Ryukyu subduction zone for Taiwan region

**Authors:** Yu-Sheng Sun, Po-Fei Chen, Chien-Chih Chen, Ya-Ting Lee, Kuo-Fong Ma, Tso-Ren Wu

**Key:** 9C78-0668-60FF-F9C0-81B7

This certificate may be verified at **secure.authorservices.springernature.com/certificate/verify**.

American Journal Experts provides a range of editing, translation and manuscript services for researchers and publishers around the world. Our top-quality PhD editors are all native English speakers from America's top universities. Our editors come from nearly every research field and possess the highest qualifications to edit research manuscripts written by non-native English speakers. For more information about our company, services and partner discounts, please visit www.aje.com.

**Response (in black) to the comments of Reviewers (in blue)**

Reviewer #1:

In Table 1, if at hand please add the water depth at the tide gauges as they appear in the computational mesh. This value is needed to reproduce the results.

We have added the values of water depth in Table 1. [Pages 26-27]

Reviewer #2:

Page 1, Abstract, line 9. I suggest to change, "tsunami earthquakes" by "tsunamigenic earthquakes", to avoid any confusion with the "tsunami-earthquake" itself. I assume authors refer to any kind of earthquake that generates a tsunami (e.g. regular earthquakes, tsunami-earthquakes, etc.).

We have done it. [Page 1, line 12]

Page 3, line 16. Please, insert "vary" after (Delta_sigma).

We have done it. [Page 4, line 2]

Page 3, line 21. Change "is" by " are".

We have done it. [Page 4, line 7]

Page 3, line 22. I suggest to change "a definite" by "the assumed".

We have done it. [Page 4, line 8]

Page 3, line 27. I think a word is missing in the sentence "..can be transformed magnitude Mw", so, I will suggest, "...can be transformed to magnitude Mw".

Thank you. We have done it. [Page 4, lines 14-15]

According to the suggestion of English editing, it was written "…can be transformed into the magnitude $M_w$".

Page 4, line 2. Please, insert the physical unit in the value of M0, I guess it is [dyne-cm].

We have done it. [Page 4, line 17]

Page 4, line 6. For better description, complete the word "temporal" by "spatio-temporal".

We have done it. [Page 4, line 21]

Page 4, line 13. I suggest to insert "The" before "k-2".
We have done it. [Page 5, line 2]

Page 4, line 15. For a better reading, I will suggest to change "self-similar introducing the...", "self similarity introduced the...".
We have done it. [Page 5, lines 3-4]
According to the suggestion of English editing, we modified the sentence.

Page 4, line 20. I think instead of "convolution in the Fourier domain" it should be, "multiplication in the Fourier domain", because the 2D Fourier spectrum of the random realization of slip is multiplied by k-2 in the Fourier domain beyond some characteristic wavelength.
We have done it. [Page 5, line 9]
In Lavallée and Archuleta, (2003) and Lavallée et al. (2006), they both used "convolution" to describe this calculation, but we agree that using multiplication is more appropriate.

Page 4, line 25. I suggest to replace "4" by "four".
We have done it. [Page 5, line 14]

Page 5, line 6. The "convoluting" operation is not correct. I will suggest to write something like, "by imposing a self-similar characteristic...".
We have done it. [Page 5, lines 24-25]

Page 5, line 8. Correct "gird" by "grid".
We have done it. [Page 5, lines 26-27]

Page 5, line 10. I will suggest to insert "shown in" before "Figure 1a".
We have done it. [Page 5, line 29]

Page 5, line 13. To complete the idea, I suggest to insert "faulting" before "mechanism".
We have done it. [Page 6, line 3]

Page 5, line 13-14. I think the sentence "In addition, the inversed slip distribution in study region is lack to do the analysis of Levy PDF" could be better executed. For instance, "There are not inverted slip models of past earthquakes in the study area to do the analysis of Levy PDF parameters.", or something like that.
We have done it. [Page 6, lines 4-5]

Page 5, line 23. I will suggest to insert "the plate interface" before "..is locked..".
We have done it. [Page 6, line 17]

Page 5, line 24. To complete the idea, I suggest to add "over the whole fault plane" after "...uniform slip distribution.".
We have done it. [Page 6, lines 21-22]

Page 7, line 27. Please, provide physical units to 1.024, m ?
We have done it. [Page 8, line 30]

Page 9, line 5. It is just a suggestion, but to better precise the idea, I suggest to modify the phrase "...is parallel the subduction zone.." by "..is parallel to the trench axis of the subduction zone", or something like that.
We have done it. [Page 10, lines 21-22]

Page 9, line 6. Insert "along" before "these".
We have done it. [Page 10, line 26]

Page 9, Paragraph 2 and 3. To help the reader, I suggest to label the four NPPs (Nuclear Power Plants) in Figure 5, map on the right. The NPPs are labeled in Figure 2, but because authors discuss the NPP3, NPP2, etc, with respect to the PTA at different locations along the coast of Taiwan in Figure 5 again, it will be useful to see the label of each NPP in this figure.
We have done it. [Page 24]

Page 9, line 6. Wildest ?, or should it be "widest" ?. Please, clarify.
We have done it. [Page 13, line 6]
Thank you. I find this word on page 11, the section of Conclusion.

Page 9, line 7. Please, provide physical units to 1.63, m ?.
We have done it. [Page 13, line 7]
* * *
Figures
* * *
Figure 2. I will suggest to complement "(5x5 km2)", by "(5x5 km2 grid size)".
We have done it. [Page 19, line 5]

Figure 5. See my comments above (Page 9, Paragraph 2 and 3). It will be useful to label each NPP in the map on the right. Please, describe a little bit the map on the right in the caption. For instance, "Map of Taiwan with station locations and four NPP (yellow squares)."
We have done it. [Page 24]

p.s.
Figure 1.
We have changed "Fit:" to "fit".

Figure 2.
We modified the resolution.

Figure 4.
We changed "hour" to "hours" for x-label.

Figure 6.
We changed "hight" to "height" for y-label.

[revised manuscript text omitted]